# Atomic structures of fibrillar segments of hIAPP suggest tightly mated β-sheets are important for cytotoxicity

**Pascal Krotee[1,2,3,4]*, Jose A Rodriguez[1,2,4], Michael R Sawaya[1,2,4], Duilio Cascio[1,2,4], Francis E Reyes[5], Dan Shi[5], Johan Hattne[5], Brent L Nannenga[5], Marie E Oskarsson[6], Stephan Philipp[7], Sarah Griner[1,2,4], Lin Jiang[3,8,9], Charles G Glabe[7,10], Gunilla T Westermark[6], Tamir Gonen[5], David S Eisenberg[1,2,3,4]***

[1]Department of Biological Chemistry, Howard Hughes Medical Institute, University of California, Los Angeles, Los Angeles, United States; [2]Department of Chemistry and Biochemistry, University of California, Los Angeles, Los Angeles, United States; [3]Molecular Biology Institute, University of California, Los Angeles, Los Angeles, United States; [4]UCLA-DOE Institute, University of California, Los Angeles, Los Angeles, United States; [5]Janelia Research Campus, Howard Hughes Medical Institute, Ashburn, United States; [6]Department of Medical Cell Biology, Uppsala University, Uppsala, Sweden; [7]Department of Molecular Biology and Biochemistry, University of California, Irvine, Irvine, United States; [8]Department of Neurology, David Geffen School of Medicine, University of California, Los Angeles, Los Angeles, United States; [9]Brain Research Institute (BRI), University of California, Los Angeles, Los Angeles, United States; [10]Biochemistry Department, Faculty of Science and Experimental Biochemistry Unit, King Fahd Medical Research Center, King Abdulaziz University, Jeddah, Saudi Arabia

**Abstract** hIAPP fibrils are associated with Type-II Diabetes, but the link of hIAPP structure to islet cell death remains elusive. Here we observe that hIAPP fibrils are cytotoxic to cultured pancreatic β-cells, leading us to determine the structure and cytotoxicity of protein segments composing the amyloid spine of hIAPP. Using the cryoEM method MicroED, we discover that one segment, 19–29 S20G, forms pairs of β-sheets mated by a dry interface that share structural features with and are similarly cytotoxic to full-length hIAPP fibrils. In contrast, a second segment, 15–25 WT, forms non-toxic labile β-sheets. These segments possess different structures and cytotoxic effects, however, both can seed full-length hIAPP, and cause hIAPP to take on the cytotoxic and structural features of that segment. These results suggest that protein segment structures represent polymorphs of their parent protein and that segment 19–29 S20G may serve as a model for the toxic spine of hIAPP.

***For correspondence:** pkrotee@ucla.edu (PK); david@mbi.ucla.edu (DSE)

**Competing interests:** The authors declare that no competing interests exist.






## Introduction

Amyloid fibrils are associated with more than 25 diseases, including Alzheimer's disease, Parkinson's disease, and Type-II Diabetes (T2D) (***Eisenberg and Jucker, 2012***). The fibrils observed in each disease are composed of a particular protein; in T2D, amyloid fibrils are composed of human islet amyloid polypeptide (hIAPP) (***Westermark et al., 1987***; ***Cooper et al., 1988***). hIAPP is a 37 residue

**eLife digest** In Type-II Diabetes, an individual's cells fail to respond correctly to the hormone insulin, leaving them unable to counteract high levels of sugar in the blood. Another hormone, human islet amyloid polypeptide (hIAPP), works with insulin to regulate blood sugar levels. hIAPP is an amyloid protein, which means that it can lose its normal structure and form fibrils. Fibrils are difficult for cells to break down and are often associated with disease. Indeed, fibrils of hIAPP often form in the pancreas as part of Type-II Diabetes.

Some studies have shown that hIAPP fibrils are toxic to pancreatic cells and worsen the symptoms of Type-II Diabetes. Others suggest that it is the process of fibril formation that is toxic, not the fibrils themselves. Although the structures of the fibrils have been described, whether these structures cause cell toxicity has not been investigated.

Krotee et al. have now explored the structures of two overlapping segments of hIAPP using a new cryo electron microscopy method called MicroED that is ideal for studying such segments. One segment, called 19-29 S20G, forms a standard amyloid fibril structure that is similar to the structure of full-length hIAPP fibrils. Adding these segments to human cells causes similar levels of toxicity as the full-length hIAPP fibrils. The second segment, called 15-25 WT, forms a non-toxic structure that is less stable than standard amyloid fibrils.

The results presented by Krotee et al. support the view that standard amyloid fibril structures are toxic to cells and suggest that 19-29 S20G may be a good model to use when studying how full-length hIAPP fibrils behave. The structure of 19-29 S20G may also be useful as a template for designing molecules that block amyloid fibril growth. If amyloid fibrils cause cell toxicity in the pancreas, then these molecules could be used to treat Type-II Diabetes.


polypeptide hormone that is co-secreted with insulin to modulate glucose levels (***Roberts et al., 1989***; ***Westermark et al., 2011***).

Researchers have accumulated substantial evidence for a correlation between hIAPP aggregation and pancreatic β-cell death in the course of the disease, T2D. Approximately 90% of pancreatic tissue samples taken post-mortem from T2D patients contain islet amyloid primarily composed of hIAPP (***Höppener et al., 2000***). The extent of islet amyloid positively correlates with pancreatic β-cell loss and insulin dependence (***Maloy et al., 1981***; ***Esapa et al., 2005***; ***Jurgens et al., 2011***). Additional support for a link comes from comparison of human and mouse IAPP: mouse IAPP differs from human IAPP by only six residues, 3 of which are β-strand breaking prolines. Consequently, mouse IAPP does not aggregate (***Nishi et al., 1989***; ***Westermark et al., 1990***). Moreover, mice can be induced to develop islet amyloid and T2D when they are engineered to express human IAPP and fed a high fat diet (***Verchere et al., 1996***; ***Westermark et al., 2000***). Perhaps the strongest support for a link is the mutation in hIAPP, hIAPP-S20G; segments that contain this mutation aggregate more quickly, contribute to increased pancreatic β-cell apoptosis, and are associated with early onset T2D in families who carry this lesion (***Sakagashira et al., 2000***; ***Cao et al., 2012***; ***Meier et al., 2016***; ***Sakagashira et al., 1996***; ***Lee et al., 2001***; ***Morita et al., 2011***).

Although a link between hIAPP aggregation and pancreatic β-cell death is well established, precisely which type of hIAPP aggregate contributes to pancreatic β-cell death and insulin dependence remains undetermined. Using mostly in vitro studies, researchers have presented evidence for toxicity of multiple types of hIAPP aggregates. Early studies suggest that amyloid fibrils are the primary cytotoxic species because preparations that contain fibrillar hIAPP were more cytotoxic than soluble preparations of the protein (***Lorenzo et al., 1994***; ***Lorenzo and Yankner, 1994***; ***Schubert et al., 1995***; ***Kapurniotu, 2001***). Using cells and transgenic rodents as disease models, other studies found hIAPP fibrils to be associated with apoptosis, β-cell loss, and T2D severity (***O'Brien et al., 1995***; ***Hiddinga and Eberhardt, 1999***; ***Janson et al., 1996***; ***Hull et al., 2005a***, ***2005b***; ***Pilkington et al., 2016***). In contrast, some studies show that the process of hIAPP fibril formation, not the amyloid fibrils themselves, is the source of toxicity (***Schlamadinger and Miranker, 2014***; ***Oskarsson et al., 2015***). However, most current research studies suggest soluble pre-fibrillar oligomers are the primary type of toxic aggregate. Support for oligomers as the primary cytotoxic species comes from

the observation of oligomers associated with caspase activity and ER stress, which precede the formation of extracellular amyloid fibrils (**Meier et al., 2006**; **Ritzel et al., 2007**; **Bram et al., 2014**; **Mukherjee et al., 2015**; **Lin et al., 2007**; **Huang et al., 2007**; **Haataja et al., 2008**; **Abedini et al., 2016**). Several recent studies show that hIAPP fibrils are relatively inert and do not exert obvious toxicity. Despite these extensive in vitro studies, it is not clear that the toxic aggregates they describe also elicit toxicity in vivo.

In closer agreement with earlier studies, we find that hIAPP preparations that contain fibrils are cytotoxic to a rat pancreatic β-cell line, thus motivating us to determine the structure of the spine of hIAPP fibrils. If fibrils are a bona fide type of toxic aggregate in vivo, then determining the atomic structure of the spine of hIAPP fibrils is a logical approach for advancing our understanding of disease-relevant targets (**Wiltzius et al., 2008**, **2009a**; **Soriaga et al., 2016**). Furthermore, we can utilize atomic structures as templates for structure-based design of novel therapeutics that may protect against pancreatic β-cell death. Although full-length amyloid proteins have so far been resistant to crystallization, select protein segments that form the spines of amyloid fibrils do form crystals (**Nelson et al., 2005**; **Sawaya et al., 2007**; **Rodriguez et al., 2015**). Indeed, the atomic structures of nearly 90 amyloid spines have been revealed in this manner. Other studies have taken an alternative approach: they employed solid-state NMR to gain detailed structural insights into hIAPP fibril structure (**Luca et al., 2007**; **Weirich et al., 2016**); some of these structures have spurred successful inhibitor designs (**Mirecka et al., 2016**). Here, we use the cryoEM method MicroED to determine the atomic structure of two 11-residue segments, termed spine segments, that span the amyloid spine of hIAPP.

# Results

## hIAPP preparations that contain fibrils are cytotoxic to cultured rat pancreatic β-cells

To compare the cytotoxic effects of oligomeric and fibrillar hIAPP, we generated hIAPP preparations that contained either amyloid oligomers or fibrils. We did this by aging the same concentration of hIAPP for 0 and 24 h time periods. Aging hIAPP for 24 h yielded amyloid fibrils and no detectable oligomers as assessed by Thioflavin-T (ThT) binding, negative-stain transmission electron microscopy (TEM), and a dot blot assay using the fibrillar oligomer-sensitive antibody, LOC (**Figure 1A**). Aging hIAPP for 0 h, which is a freshly dissolved hIAPP sample, yielded oligomers as assessed by a dot blot assay using LOC, and no amyloid fibrils (**Figure 1—figure supplement 1A**). Of note, we probed both hIAPP preparations with 25 different conformational antibodies that are known to bind soluble oligomers, but only LOC showed binding to any of our preparations. Although LOC was raised against hIAPP fibrils (**Kayed et al., 2007**), studies show that it also recognizes fibrillar oligomers (**Wu et al., 2010**), which share structural epitopes with amyloid fibrils and are structurally distinct from A11-positive pre-fibrillar oligomers.

We observe that hIAPP preparations that contain fibrils are significantly more cytotoxic to rat pancreatic β-cells than hIAPP preparations that contain oligomers but no detectable fibrils (**Figure 1B and C**). We assayed the cytotoxicity of the hIAPP preparations to Rin5F cells, a rat pancreatic β-cell line (**Gazdar et al., 1980**) using two metrics: 3-(4,5-dimethylthiazol-2-yl)−2,5-diphenyltetrazolium bromide (MTT) dye reduction, an indicator of good metabolic health (**Mosmann, 1983**; **Liu et al., 1997**), and activation of caspase-3/7, an indicator of apoptosis (**Budihardjo et al., 1999**). Furthermore, the insoluble fraction of the hIAPP 24 h sample, which contains fibrils (**Figure 1—figure supplement 1B**), is cytotoxic, while the soluble fraction is not (**Figure 1D**), further suggesting that fibrils are the toxic aggregate in our studies.

Although we do not detect oligomers in the 24 h sample, we cannot rule out the possibility that it may contain some undetectable population of slowly forming, yet highly toxic oligomers that associate with fibrils. Despite this possibility, we chose to focus on studying fibrillar structures of hIAPP.

## Selection of amyloid spine segments for structural studies

Given that hIAPP fibrils are cytotoxic, we sought to identify the residues that compose their amyloid spine. We identified residues 15–29 as the amyloid spine based on several lines of evidence and previous work by others (**Westermark et al., 1990**; **Moriarty and Raleigh, 1999**; **Goldsbury et al.,**

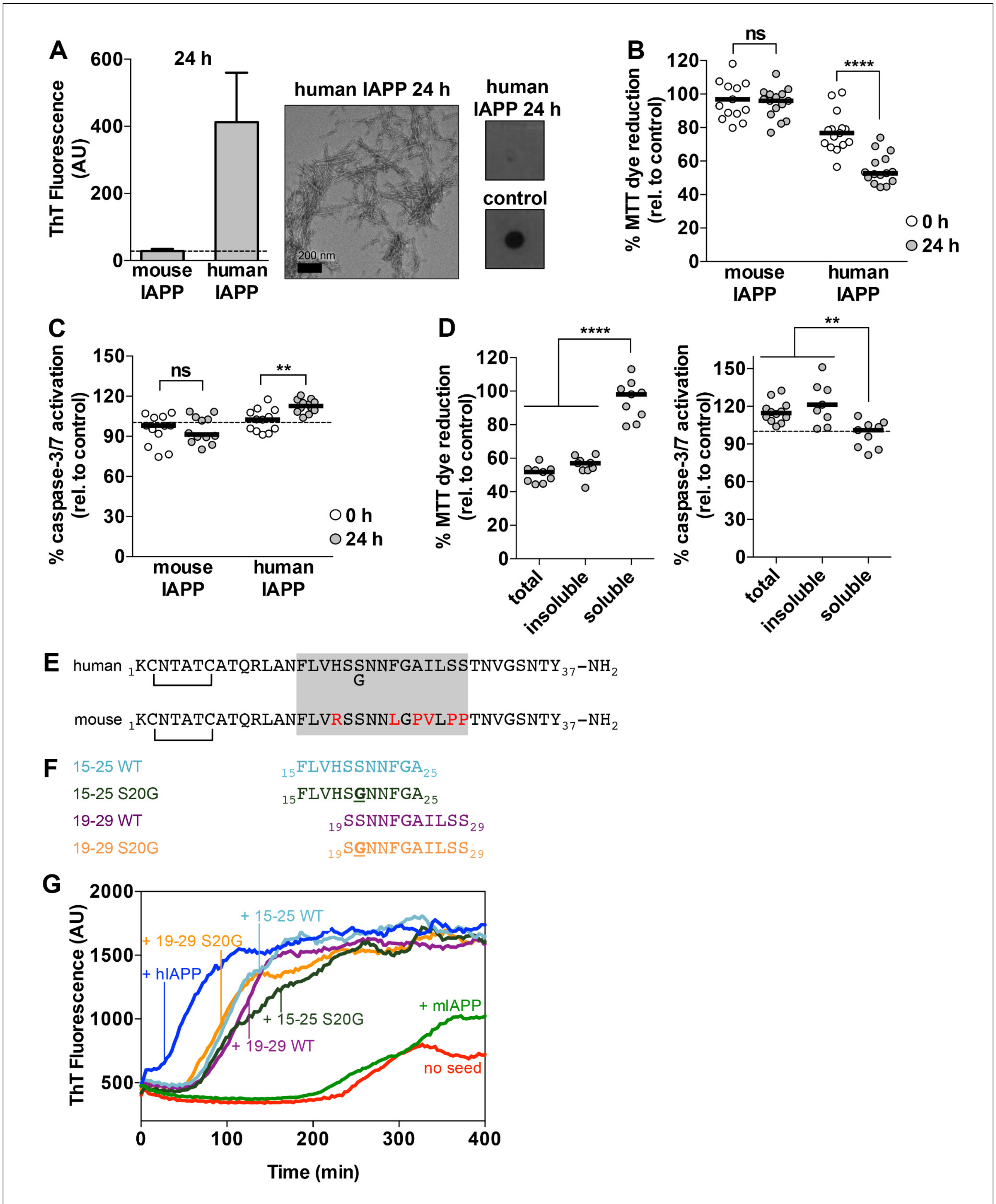


**Figure 1.** Preparations of hIAPP that contain amyloid fibrils are cytotoxic to a rat pancreatic β-cell line. (A) Human IAPP (hIAPP) aged for 24 h contains amyloid fibrils and no detectable oligomers. Amyloid fibrils were observed using ThT binding and TEM. Oligomers were detected using a dot bot assay with the polyclonal anti-oligomer antibody, LOC. hIAPP oligomers were used as the positive control for LOC binding. The dashed line on the ThT binding graph indicates ThT fluorescence of vehicle alone. (B) and (C) hIAPP aged for 24 h is significantly more cytotoxic than hIAPP aged for 0 h. In

*Figure 1 continued*

these experiments, 50 μM human and mouse IAPP were aged for the designated time periods and then they were applied to cells at 5 μM final concentration. Mouse IAPP (mIAPP), which does not form amyloid fibrils, is not cytotoxic regardless of time period of aging. Black horizontal bars indicate the median (n = 12–15 across 4–5 biological replicates, each with three technical replicates). (**B**) Rin5F cells treated with hIAPP aged for 24 h reduce significantly less MTT dye than Rin5F cells treated with hIAPP aged for 0 h (ns=not significant; ****p<0.0001 using an unpaired t-test with equal standard deviations). (**C**) Rin5F cells treated with hIAPP aged for 24 h exhibit significantly higher caspase-3/7 activation than Rin5F cells treated with hIAPP aged for 0 h. Additionally, Rin5F cells treated with hIAPP aged for 24 h exhibit significantly higher caspase-3/7 activation than vehicle-treated cells (***p=0.0008 using an ordinary one-way ANOVA), but Rin5F cells treated with hIAPP aged for 0 h do not (p=0.4286 using an ordinary one-way ANOVA) (ns=not significant; **p=0.0011 using an unpaired t-test with equal standard deviations). (**D**) The insoluble fraction of hIAPP aged 24 h, which contains amyloid fibrils and no detectable oligomers, contains the cytotoxic species. Cytotoxicity was measured using MTT dye reduction and detection of caspase-3/7 activation (****p<0.0001; **p<0.0013; n = 9 across three biological replicates, each with three technical replicates). (**E**) Amino acid sequences of human IAPP and mouse IAPP. The location of the early onset familial mutation, S20G, is shown below the human sequence. Red residues in the mouse sequence differ from the human sequence. The amyloid spine of human IAPP and the corresponding region in the mouse sequence is enclosed in the gray box. (**F**). Schematic of protein segments that span the amyloid spine, hereon referred to as spine segments, targeted for characterization. (**G**) Fibrils of spine segments seed hIAPP fibril formation, suggesting that spine segments embody structural characteristics of full-length hIAPP fibrils. 10 μM hIAPP was seeded with 10% (v/v) monomer equivalent of pre-formed, unsonicated seed of each spine segment. mIAPP, which does not contain amyloid fibrils, does not seed hIAPP fibril formation. Curves show average of 4 technical replicates.


The following figure supplements are available for figure 1:

**Figure supplement 1.** Characterization of hIAPP aged for 0 h and the soluble and insoluble fractions of hIAPP aged for 24 h.


**Figure supplement 2.** All spine segments form amyloid fibrils or 3D crystals only a few hundred nanometers thick, as observed using TEM.


**Figure supplement 3.** Technical replicates and control samples for ThT assay in ***Figure 1G***.


***2000***; ***Tenidis et al., 2000***). First, the sequence of mouse IAPP (mIAPP), which is non-amyloidogenic, differs from human IAPP only within this region (***Figure 1E***). Second, the only known familial disease mutation in hIAPP, hIAPP-S20G, also occurs within this region (***Figure 1E***) (***Sakagashira et al., 2000***; ***Cao et al., 2012***). Third, previous work by our laboratory has shown that Phe15 may be part of the amyloid spine because it is required for stabilizing an on-pathway α-helical dimer and mutating this residue can delay fibril formation (***Wiltzius et al., 2009b***).

For these reasons, we chose to focus on two overlapping 11-residue segments within this region of the sequence: residues 19–29 and residues 15–25. We chose to study the WT and early onset S20G mutation segments (***Figure 1F***). All four spine segments form amyloid fibrils or crystals (***Figure 1—figure supplement 2***) that seed full-length hIAPP fibril formation (***Figure 1G***, ***Figure 1—figure supplement 3***), suggesting that the spine segments embody structural characteristics of full-length hIAPP fibrils.

## Segment 19–29 S20G forms pairs of β-sheets tightly mated by a dry interface

To determine the structure of segment 19–29 S20G, we used Micro-Electron Diffraction (MicroED). MicroED employs a standard cryo electron microscope (cryoEM) in diffraction mode for data collection from 3D crystals only a few hundred nanometers thick (***Figure 2A***; ***Figure 3A***) (***Shi et al., 2013***; ***Nannenga et al., 2014a***, ***2014b***; ***Hattne et al., 2015***; ***Liu et al., 2016***). Such thin crystals are capable of producing measurable Bragg peaks because electrons interact with matter more strongly than X-rays. Indeed, we found that the nano-sized 3D crystals used for MicroED produced higher resolution diffraction than relatively larger crystals suited for structure determination at a microfocus X-ray beamline (***Figure 2A***). Evidently, micron-thick needle crystals are sufficient for X-ray structure determination with six or seven residue peptides, but not for 11-residue peptides. These experiences closely mirrored those in the determination of the atomic structure of the toxic core of α-synuclein (***Rodriguez et al., 2015***), an 11-residue segment that forms the spine of amyloid fibrils associated with Parkinson's disease.

The MicroED atomic structure of segment 19–29 S20G reveals pairs of parallel in-register β-sheets mated face-to-face by interdigitation of side-chains and exclusion of water molecules (***Figure 3B and C***;***Table 1***). This arrangement is termed a class I steric-zipper. Such features are observed for

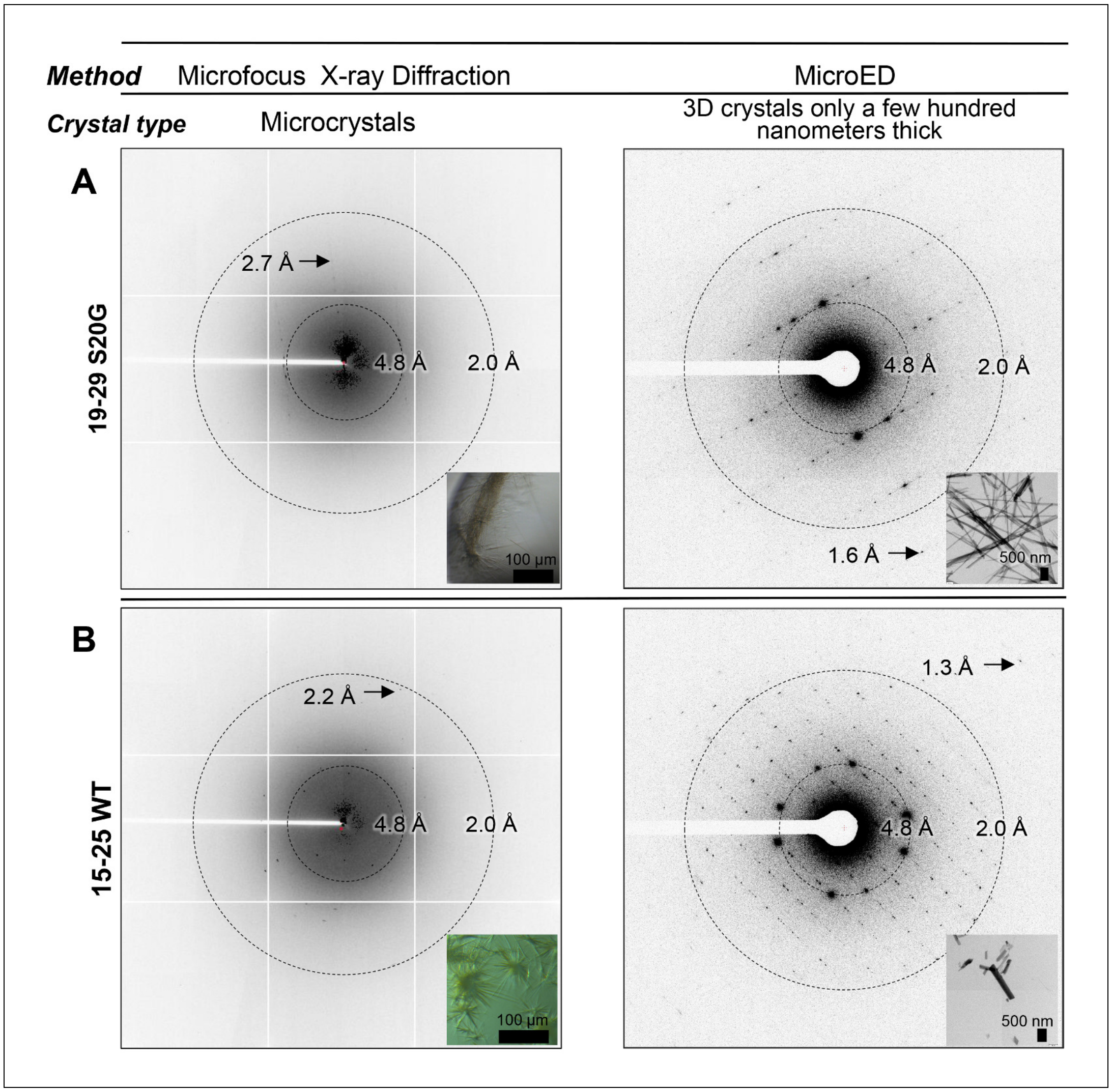


**Figure 2.** Bragg peaks produced by MicroED from 3D crystals only a few hundred nanometers thick are observed at higher resolution than peaks produced by X-ray diffraction at a microfocus beamline from microcrystals 10,000 times larger. (**A**) 3D crystals of 19–29 S20G (right, inset) diffract to 1.6 Å using MicroED, a whole angstrom better resolution than the microcrystals of 19–29 S20G (left, inset). (**B**) 3D crystals of 15–25 WT (right, inset) diffract to 1.4 Å using MicroED, whereas microcrystals of 15–25 WT diffract to 2.2 Å using Microfocus X-rays (left, inset).

amyloid spines of other proteins and have been associated with pathology (***Nelson et al., 2005***; ***Sawaya et al., 2007***; ***Ivanova et al., 2009***; ***Colletier et al., 2011***; ***Liu et al., 2011***). This zipper contains a tightly packed hydrophobic core consisting of Phe23, Ala25, and Ile27. Phe23 is the central and largest contributor the hydrophobic core, consistent with multiple other experiments (***Tenidis et al., 2000***; ***Griffiths et al., 1995***; ***Jack et al., 2006***; ***Madine et al., 2008***). The dry interface buries 265 Å$^2$ of surface area per strand, which equates to 24 Å$^2$ per residue. This interface is one of the largest and most complementary of any structurally determined steric-zipper interface (***Supplementary file 1***); it has a shape complementary of 0.85. The dry interface is nearly as large as the toxic core of α-synuclein (***Rodriguez et al., 2015***), but with higher shape complementarity.

The β-sheets of the 19–29 S20G atomic structure possess a curvature that is not common in shorter hIAPP protein segments (***Wiltzius et al., 2008***, ***2009a***; ***Soriaga et al., 2015***). To assess β-sheet curvature, we compared the root mean square deviations (RMSD's) of sheets from planarity across all hIAPP protein segment atomic structures determined to date (***Supplementary file 1***). The 19–29 S20G structure ranks in the upper half of the list (***Figure 3—figure supplement 2***), containing

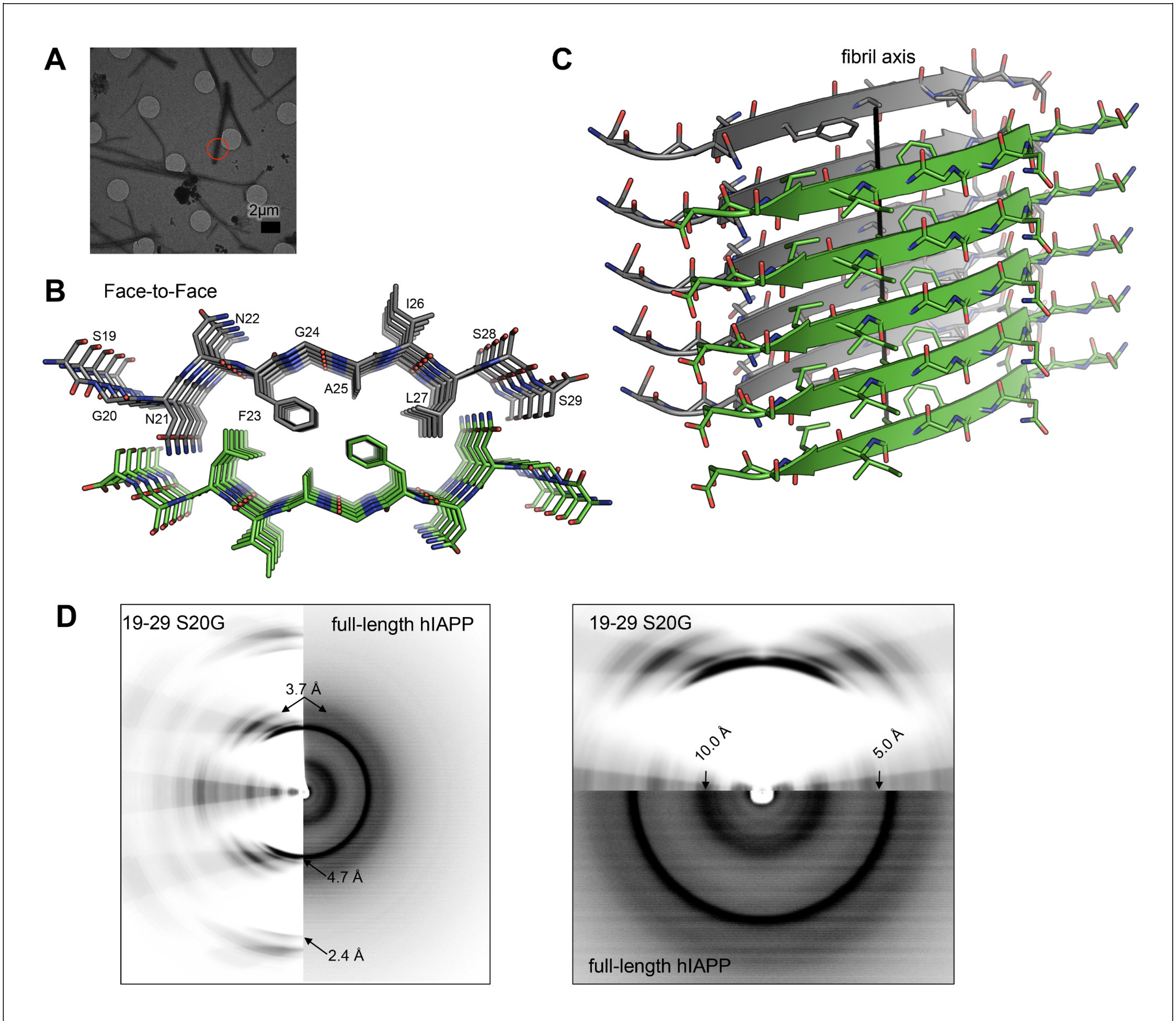


**Figure 3.** The MicroED atomic structure of segment 19–29 S20G reveals pairs of β-sheets mated by a dry interface. (**A**) Electron micrograph of 3D crystals used for data collection. The red circle represents the area of the crystal used for diffraction. (**B**) Pairs of β-sheets are oriented face-to-face and they are tightly mated by a dry interface that excludes water. The dry interface is formed by tightly packed, interdigitating side-chains. This panel shows 5 β-strands or layers along the 'a' dimension of the unit cell; the average crystal used for data collection is 10,400 layers long in the 'a' dimension. (**C**) Orthogonal view of the steric-zipper formed by the dry interface. (**D**) The similarity between the fiber diffraction pattern calculated from the structure shown in Panel C and the fiber diffraction observed from full-length hIAPP fibrils supports the dry interface as a model for the amyloid spine of full-length hIAPP fibrils. Along the meridian (left panel), the dry interface and full-length hIAPP fibrils share reflections at 4.7 Å and and 2.4 Å (black arrows). Additionally, along the off-meridonal, the diffraction patterns share a reflection at 3.7 Å. It is difficult to see the reflection at 2.4 Å in the full-length hIAPP fiber diffraction image, but the reflection is clearly visible in the radial profile in ***Figure 3—figure supplement 1***. Along the equator (right panel), the dry interface and full-length hIAPP fibrils share reflections at 10.0 Å and 5.0 Å (black arrows). The right panel is magnified 2X to more clearly show the low-resolution reflections along the equator.


The following figure supplements are available for figure 3:

**Figure supplement 1.** The crystal packing of segment 19–29 S20G reveals a second interface, termed the 'Back-to-Back' or wet interface, which does not form the amyloid spine.


*Figure 3 continued*

**Figure supplement 2.** Scatter plot of sheet RMSD from planarity values for all hIAPP protein segment structures determined to date.

**Figure supplement 3.** 19-29 WT and S20G have similar fibrillar structures.

**Figure supplement 4.** Radial profile calculated from the X-ray diffraction pattern given by cytotoxic full-length hIAPP fibrils.

both sheet curvature and a sharp kink. Most of the shorter peptides are nearly flat, but some have sharp kinks. The significance of deviation from planarity is not yet clear.

The similarity between the fiber diffraction pattern calculated from this steric-zipper and the fiber diffraction pattern collected from full-length hIAPP fibrils tends to validate the 19–29 S20G atomic structure as a model for the amyloid spine of full-length hIAPP (***Figure 3D***). The diffraction patterns share several key features, including reflections at 4.7 Å and 2.4 Å along the meridian, a reflection at 3.7 Å along the off-meridian (left panel), and reflections at 10.0 Å and 5.0 Å along the equator (right panel).

Structural studies performed here and elsewhere by others suggest that 19–29 WT can form a similar dry interface to the one observed in the 19–29 S20G atomic structure. Radial profiles calculated from X-ray fiber diffraction of 19–29 WT and 19–29 S20G fibrils show strong reflections in common at 4.6 Å, 8.4 Å and 8.7 Å, and 34.7 Å, indicative of interstrand, intersheet, and proto-filament spacing, respectively (***Figure 3—figure supplement 3***). A previous study of 20–29 WT fiber diffraction revealed comparable reflections, which the authors used to formulate a fibril model of 20–29 WT that roughly agrees with our 19–29 S20G atomic structure (***Madine et al., 2008***). Our atomic structure and their model differ by a small shift in registration between sheets, allowing for tighter packing in the atomic structure. These results are consistent with earlier findings by Cao and coworkers, who observed that hIAPP-WT fibrils seed hIAPP-S20G fibril formation, thus suggesting a shared fibrillar structure (***Cao et al., 2012***).

Although the WT and mutant segments likely form similar structures, the structure of the mutant segment may be more stable. The stability of the mutant segment may stem from the early onset Gly20 mutation, which adopts an unusual geometry ($\varphi = -101.7°$ and $\psi = 107.5°$) that creates a kink in the peptide backbone. To investigate this hypothesis, we generated a model of 19–29 WT consisting of a mated pair of ten-stranded sheets. The model was identical to the 19–29 S20G atomic structure with the exception that we adjusted the backbone torsion angles of Ser20 to comply with the allowed regions of the Ramachandran plot for a non-glycine residue. We compared the energies of the WT and S20G structures after minimization with FoldIt (***Cooper et al., 2010***). The dry interfaces are nearly identical between the two segments, except near Asn21, where the altered backbone torsion angles break the canonical Asn ladder hydrogen bonding interactions with neighboring Asn21 residues within the sheet and instead, form hydrogen bonds with Ser29 from the opposing sheet. The alteration separates the pair of sheets by approximately 1.5 Å in this region, and therefore the 19–29 S20G structure has a slightly lower energy than 19–29 WT (−590 REU vs. −535 REU).

## Segment 15–25 WT forms an arrangement of labile unmated β-sheets

The atomic structure of segment 15–25 WT, also determined using MicroED (***Figure 2B***; ***Figure 4A***), shows an arrangement of unmated β-sheets composed of anti-parallel out-of-register β-strands that is uncharacteristic of pathogenic amyloid fibrils (***Figure 4B***;***Table 1***). Most pathogenic amyloid fibrils are composed of β-strands that stack perpendicular to the sheet-long axis, but the β-strands in out-of-register structures stack at an angle. Deviation of strands from the fibril perpendicular is a natural consequence of the registration shift implied by out-of-register structures. The out-of-register β-strands are stabilized by extensive hydrogen bonding. Within each sheet, the β-strands form two distinct, unequal interfaces: a stronger interface with twelve hydrogen bonds, and a weaker interface with eight hydrogen bonds (***Figure 4B***). This inequality between interfaces has been observed in previous examples of out-of-register sheets (***Soriaga et al., 2015***; ***Laganowsky et al., 2012***; ***Liu et al., 2012***; ***Yu et al., 2015***). A view down the 'proto-fibril axis' of the crystal shows that the faces of adjacent sheets are wet and overlap only partially (***Figure 4C***); the asymmetric unit contains density for seven ordered water molecules and one thiocyanate molecule. The area buried between

**Table 1.** Statistics of MicroED data collection and atomic refinement.

| Sample | 19–29 S20G | 15–25 WT |
|---|---|---|
| Excitation Voltage (kV) | 200 | 200 |
| Electron Source | field emission gun | field emission gun |
| Wavelength (Å) | 0.0251 | 0.0251 |
| Total dose per crystal ($e^{-}$/ Å$^2$) | 3.4 | 2.9 |
| Frame rate (frame/s) | 0.3–0.5 | 0.3–0.5 |
| Rotation rate (°/s) | 0.3 | 0.3 |
| # crystals used | 6 | 6 |
| Total angular rotation collected (°) | 68 | 68 |
| **Merging Statistics** | 19–29 S20G | 15–25 WT |
| space group | $P2_12_12_1$ | P1 |
| Unit cell dimensions | | |
| a, b, c (Å) | 4.78, 18.6, 70.8 | 11.68, 18.18, 19.93 |
| α, β, γ (°) | 90, 90, 90 | 62.8, 88.9, 87.6 |
| Resolution (Å) | 1.9 | 1.4 |
| $R_{merge}$ | 10.6% (15.0%) | 19.9% (50%) |
| # of reflections | 1380 (221) | 9014 (153) |
| Unique reflections | 548 (115) | 2180 (84) |
| Completeness | 83% (65%) | 75% (35.3%) |
| Multiplicity | 2.5 (1.9) | 4.1 (1.8) |
| I/σ | 5.65 (3.65) | 4.33 (1.10) |
| $CC_{1/2}$ (***Diederichs, 2013***) | 98.9% | 98.5% |
| **Refinement Statistics** | 19–29 S20G | 15–25 WT |
| Reflections in working set | 546 | 2177 |
| Reflections in test set | 53 | 218 |
| $R_{work}^{\dagger}$ | 22.75% | 22.47% |
| $R_{free}$ | 27.49% | 25.90% |
| RMSD bonds (Å) | 0.01 | 0.008 |
| RMSD angles (°) | 1.2 | 1.2 |
| Ramachandran (%)‡ | | |
| Favored | 100 | 100 |
| Allowed | 0 | 0 |
| Outliers | 0 | 0 |
| PDB ID code | 5KNZ | 5KO0 |
| EMDB ID code | EMD-8272 | EMD-8273 |

*Highest resolution shell shown in parenthesis.

†$Rfactor = 100x\sum||F_{obs}| - |F_{calc}||/\sum|F_{obs}|$

$F_{calc}$ and $F_{obs}$ are the calculated and observed structure factor amplitudes, respectively. $R_{work}$ refers to the $R_{factor}$ for the data utilized in the refinement and $R_{free}$ refers to the $R_{factor}$ for 10% of the reflections randomly chosen that were excluded from the refinement.

‡Percentage of residues in Ramachandran plot regions were determined using Molprobity (***Chen et al., 2010***).


adjacent sheets is small (10.7 Å$^2$ per residue) compared to the average steric-zipper (20.1 Å$^2$ per residue). Hence, there is no dry interface between adjacent sheets in the crystal, and the structure seems labile compared to that of 19–29 S20G.

Consistent with our observation of unmated β-sheets in the atomic structure, we observe that 15–25 WT fibrils are relatively weak and reversible compared to 19–29 S20G fibrils, which possess a canonical pathogenic amyloid fibril architecture. Turbidity readings followed by negative-stain EM reveal that 15–25 WT fibrils completely disaggregate in the presence of heat and 1% SDS, but 19–29 S20G fibrils remain intact in up to 2% SDS (***Figure 4—figure supplement 1***).

Similar to 19–29 S20G, the 15–25 WT atomic structure reveals curved β-sheets. The sheets possess one of the highest RMSD's of sheets from planarity for any hIAPP protein segment structure determined to date (***Supplementary file 1***, ***Figure 4—figure supplement 2***).

X-ray fiber diffraction and radial profile analysis of 15–25 WT and 15–25 S20G fibrils indicate they form structures similar to each other (***Figure 4—figure supplement 3***). Taken together with the X-ray fiber diffraction data from the 19–29 segments, we conclude that the early onset S20G mutation does not confer a fibril morphology distinguishable from wild-type.

## Structural polymorphs elicit different cytotoxic effects

Next we investigated the cytotoxic effects of the spine segments in order to determine if any of them were similarly cytotoxic to full-length hIAPP preparations. Although the cytotoxic mechanism of hIAPP is not fully understood, several reports show hIAPP induces mitochondrial dysfunction, alters cell metabolism, and initiates activation of pro-apoptotic machinery (***Butler et al., 2003***; ***Mulder and Ling, 2009***; ***Zraika et al., 2010***; ***Magzoub and Miranker, 2012***; ***Tomasello et al., 2014***). Based on these findings, we tested the cytotoxicity of the spine segments using MTT dye reduction (***Mosmann, 1983***; ***Liu and Schubert, 1997***) and a FRET-based biosensor to assay altered metabolism and pro-apoptotic machinery activation (***Paulsson et al., 2008***), respectively.

Using MTT dye reduction, we observe that the labile 15–25 fibrils are not cytotoxic to HEK293 cells (***Figure 5A***), whereas 19–29 S20G fibrils have comparable cytotoxicity to full-length hIAPP fibrils (***Figure 5B***). To verify the cytotoxic effects of each sample, we examined the morphology of the treated cells under a light microscope. Additionally, in the context of residues 19–29, the S20G segment is significantly more cytotoxic than the WT segment, consistent with parent full-length hIAPP (***Sakagashira et al., 2000***; ***Meier et al., 2016***) (***Figure 5B***). We did not detect any oligomers present in the 15–25 WT or 19–29 S20G fibril samples using the LOC antibody (***Figure 5—figure supplement 1***).

Based on our examination of the insoluble and soluble fractions of the cytotoxic 19–29 S20G sample, we determine that the cytotoxicity of 19–29 S20G mainly resides in its fibrillar form. We tested the cytotoxicity of the total, insoluble and soluble fractions of the 19–29 S20G sample to HEK293 cells using MTT dye reduction. We observe that the insoluble fraction, which contains amyloid fibrils, is similarly cytotoxic to the total (***Figure 5C***), just as we observed with full-length hIAPP (***Figure 1D***, ***Figure 1—figure supplement 1B***). These results suggest that 19–29 S20G may form the toxic spine of full-length hIAPP.

Further evidence that 19–29 S20G may form the toxic spine of full-length hIAPP comes from our observation that (−)-epigallocatechin gallate (EGCG), a flavanol known to mitigate full-length hIAPP cytotoxicity by preventing hIAPP from forming fibrils (***Meng et al., 2010***), also mitigates 19–29 S20G cytotoxicity by preventing it from forming fibrils (***Figure 5—figure supplement 2B and C***). We hypothesize that EGCG may mitigate fibril formation of full-length hIAPP and 19–29 S20G by binding to a common site, such as the dry interface of the amyloid spine. A previous study suggested EGCG may mitigate hIAPP fibril formation by binding hIAPP via hydrophobic interactions (***Young et al., 2015***). Indeed, EGCG does not prevent fibril formation of 15–25 WT, which does not possess a dry hydrophobic interface (***Figure 5—figure supplement 2D***). In addition, these results further support our conclusion that preparations of segment 19–29 S20G that contain fibrils are cytotoxic.

Next we tested whether the spine segments activate pro-apoptotic machinery using a FRET-based biosensor assay for monitoring caspase-3 activity in real-time (***Paulsson et al., 2008***). In this assay, CHO cells are stably transfected with a construct containing enhanced cyan fluorescent protein (ECFP) and enhanced yellow fluorescent protein (EYFP) fused by a DEVD linker. FRET signal is observed by exciting ECFP at 440 nm. In cells undergoing apoptosis, active caspase-3-like proteases target and cleave the DEVD linker, resulting in loss of FRET signal. Cell viability is measured by monitoring the ratio of 540 nm/480 nm, which reports loss of FRET signal and increased caspase-3 activity.

Using this system, we observe that segment 19–29 S20G elicits the most caspase-dependent cytotoxicity of the spine segments and segments 15–25 are not cytotoxic (**Figure 5D and E**). Segment 19–29 S20G is not as cytotoxic as full-length hIAPP in this assay, possibly because hIAPP interaction with heparan sulfate proteoglycans (HSPG) is important for apoptosis induction (**Oskarsson et al., 2015**), and residues 1–8, which are missing in all of the spine segments, are required for hIAPP binding to HSPG's.

### Fibril seeds of 15–25 WT reduce the cytotoxicity of full-length hIAPP

Given that the spine segments seed full-length hIAPP fibril formation and that 19–29 S20G and 15–25 WT fibrils elicit different cytotoxic effects, we investigated whether seeding with either of the spine segments alters hIAPP cytotoxicity. To do this, we prepared seeded hIAPP at 10 µM with 10% monomer equivalent of pre-formed seeds, the same conditions used in the ThT assay in **Figure 1G**. For all cytotoxicity assays, we dilute samples 1 to 10 to the concentration specified in culture medium containing pre-plated cells. Thus, we tested the cytotoxicity of seeded hIAPP at 1 µM in order to preserve the conditions of the ThT assay.

Using MTT dye reduction, we observe that hIAPP seeded with non-toxic 15–25 WT fibrils is less cytotoxic than hIAPP alone, but hIAPP seeded with stable, toxic 19–29 S20G fibrils is similarly cytotoxic to hIAPP alone (**Figure 6**). Likewise, hIAPP seeded with stable, toxic 19–29 S20G fibrils is significantly more cytotoxic than hIAPP seeded with labile, non-toxic 15–25 WT fibrils (**Figure 6**). Seeds alone are not cytotoxic, indicating the cytotoxic effects we observe originate from the interaction of each seed with hIAPP and not the seed alone.

There are two possible explanations for the reduced cytotoxicity of the seeded 15–25 WT sample. First, the 15–25 WT seeds may seed a non-toxic species of full-length hIAPP, or second, the 15–25 WT seeds may interact with full-length hIAPP in some way that reduces its cytotoxicity.

X-ray fiber diffraction and radial profile analysis of the hIAPP fibrils used in the cytotoxicity assay reveal that fibrils formed by seeding with stable, toxic 19–29 S20G fibrils have a slightly tighter packing than fibrils formed by seeding with labile, non-toxic fibrils. hIAPP fibrils formed by seeding with stable, toxic 19–29 S20G fibrils exhibit reflections indicative of shorter equatorial Bragg spacings than hIAPP seeded with labile, non-toxic 15–25 WT fibrils (9.0 Å versus 10.0 Å) (**Figure 6—figure supplement 1**). The tighter packing of these fibrils may explain their enhanced cytotoxicity. Fiber diffraction could not be detected from seeds alone prepared under the same conditions.

## Discussion

In 1901, when Dr. Eugene Opie first observed islet amyloid in post-mortem pancreata of T2D patients, he proposed a link between the islet amyloid and T2D (**Opie, 1901**). Over a century later, multiple studies have shown an unequivocal link between hIAPP aggregation and T2D, but uncertainty remains about which type of hIAPP aggregate contributes to pancreatic β-cell death. Although most recent in vitro studies suggest soluble oligomers are the primary type of toxic aggregate, here, we find hIAPP samples that contain fibrils alter pancreatic β-cell metabolism and activate pro-apoptotic caspases.

These findings motivated us to determine the structure of the spine of hIAPP fibrils and elucidate structural features important for hIAPP cytotoxicity. To improve our likelihood of crystallization and structure determination, we selected four protein segments that span the spine. We discovered that segment 19–29 S20G forms a pair of β-sheets mated at a dry interface, a structure that shares key features with full-length hIAPP fibrils as described in the following paragraph. What's more, the fibrillar form of 19–29 S20G is cytotoxic. In contrast, segment 15–25 WT forms an unusual arrangement of single, out-of-register β-sheets that are not cytotoxic. The divergence in structure and cytotoxicity of segments 19–29 S20G and 15–25 WT suggests that strong, stable intermolecular interactions are important features of cytotoxic amyloid proteins.

The experiments of this study show that the 19–29 S20G atomic structure recapitulates many of the structural features and cytotoxic properties of hIAPP. First, preparations of 19–29 S20G that contain fibrils are cytotoxic, as is the case for full-length hIAPP. Second, X-ray fiber diffraction calculated from the dry interface of the 19–29 S20G atomic structure shares key features with fiber diffraction collected from full-length hIAPP fibrils. Third, segment 19–29 S20G elicits cytotoxicity by altering cell metabolism and activating pro-apoptotic machinery, mechanisms by which full-length hIAPP fibrils

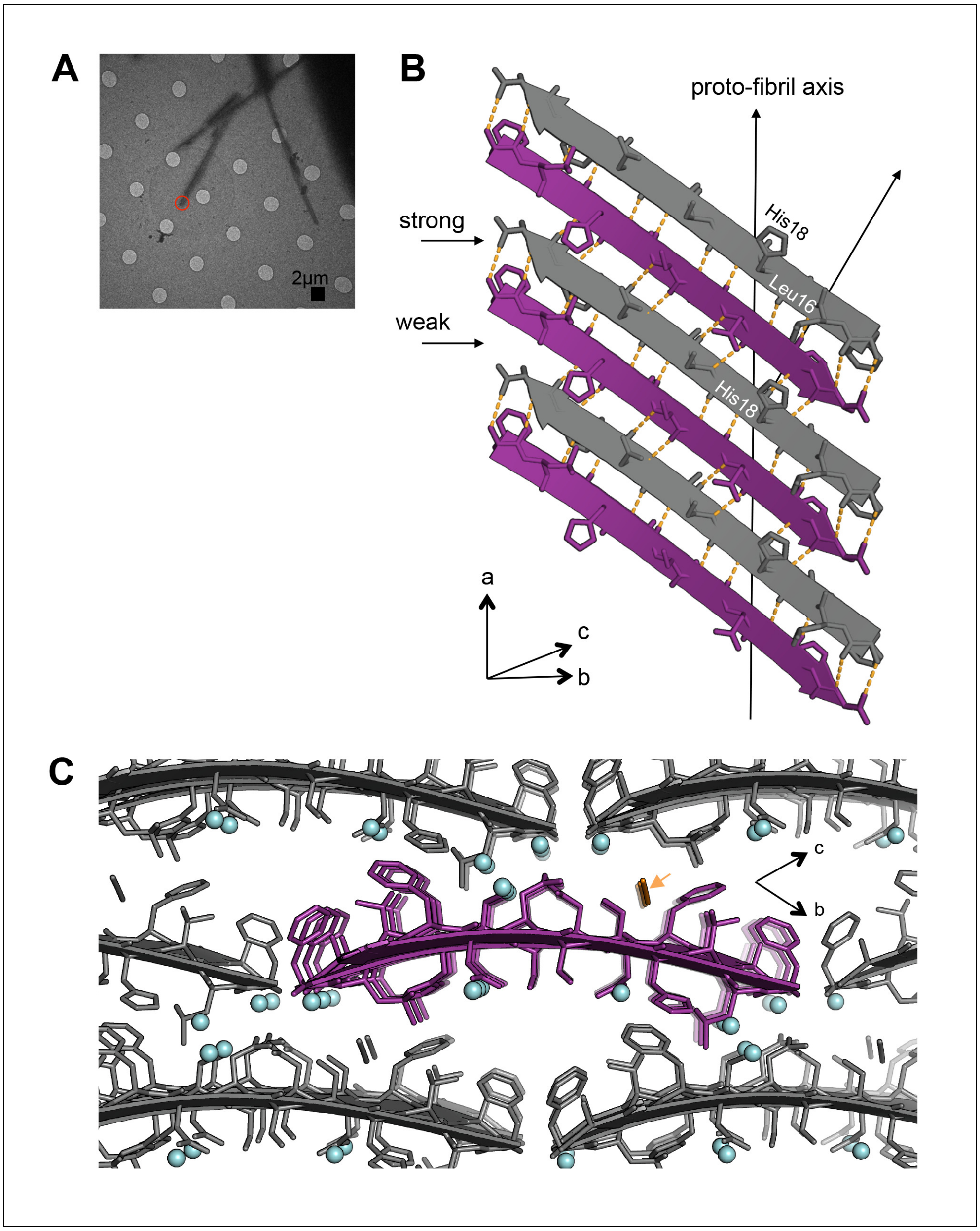


**Figure 4.** Segment 15–25 WT forms an arrangement of unmated β-sheets that is uncharacteristic of pathogenic amyloid fibrils. (**A**) Electron micrograph of 3D crystals used for data collection. The red circle represents the area of the crystal used for diffraction. (**B**) A single β-sheet contains anti-parallel out-of-register β-strands stabilized by two distinct, unequal interfaces: a stronger interface with twelve hydrogen bonds, and a weaker interface with eight hydrogen bonds. The β-strands are out-of-register by two residues because Leu16 on the first β-strand is directly above His18 on the third β-strand. (**C**) The view down the proto-fibril axis reveals hydrated interfaces between partially overlapping β-sheets. Notice that adjacent β-sheets lack side-chain interdigitation. Water molecules are shown as cyan spheres. The thiocyanate molecule is highlighted in gold in the central β-sheet and colored gray in the peripheral β-sheets.


The following figure supplements are available for figure 4:

**Figure supplement 1.** 15-25 WT fibrils are relatively weak and reversible compared to 19–29 S20G fibrils.


**Figure supplement 2.** Scatter plot of sheet RMSD from planarity values for all hIAPP protein segment structures determined to date.


**Figure supplement 3.** 15-25 WT and S20G have similar fibrillar structures.


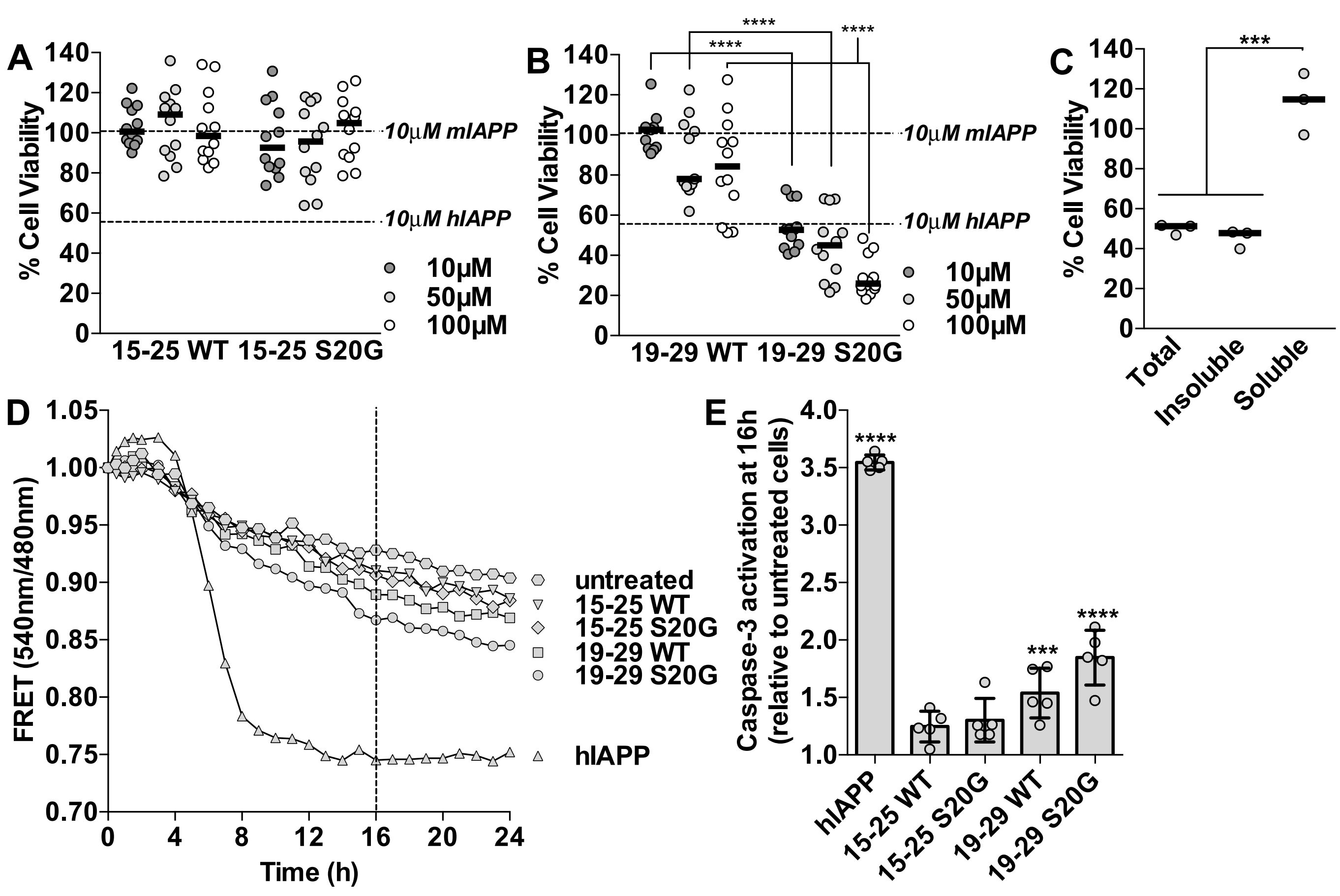


**Figure 5.** Segment 19–29 S20G forms the toxic core of hIAPP and segments 15–25 are not toxic. (**A** and **B**) Fibrils were formed by incubating the spine segments overnight under quiescent conditions, the same conditions used to prepared full-length hIAPP fibrils. Next, the samples were applied to HEK293 cells at the specified concentrationsand then cell viability was quantified using MTT dye reduction. Bars show median cell viability; dashed lines show median cell viability from 10 µM mIAPP and hIAPP. (**A**) 15–25 WT and 15–25 S20G fibrils are not toxic compared to full-length hIAPP fibrils (n = 12 across four biological replicates, each with three technical replicates). (**B**) 19–29 WT fibrils are mildly cytotoxic and 19–29 S20G fibrils are significantly more cytotoxic than 19–29 WT fibrils (****p<0.0001 using a Mann-Whitney U test; n = 12 across four biological replicates, each with three technical replicates). 19–29 S20G fibrils (10 µM) are similarly cytotoxic to full-length hIAPP fibrils at the same concentration (lower dashed line) (p=0.09 using an unpaired t-test with equal standard deviations). (**C**) The insoluble fraction of the 50 µM 19–29 S20G cytotoxic preparation contains the cytotoxic species. 19–29 S20G fibrils were formed overnight at room temperature and then pelleted by centrifugation. The soluble fraction was carefully removed and then filtered to ensure it contained no insoluble material. The insoluble material was resuspended in its original volume. Each sample was applied to HEK293 cells and then cell viability was quantified with MTT dye reduction (***p<0.0002 using an ordinary one-way ANOVA; n = 3 technical replicates) (**D**) and (**E**) Using a FRET-based biosensor assay for monitoring caspase-3 activity in real-time, 19–29 S20G fibrils induce the most caspase-3 activity, whereas segments 15–25 did not induce caspase-3 activity, consistent with the MTT dye reduction assay results. 50 µM of each spine segment seeded with 166 nM seeds was applied to stably transfected CHO cells. (**D**) Fold difference was recorded over 24 h. Datapoints represent average fold difference. The dashed line represents the 16 h mark. (**E**) Average levels of caspase-3 activation after a 16 h incubation relative to untreated cells (***p<0.0002; ****p<0.0001 using an ordinary one-way ANOVA, Bonferroni correction; n = 5 technical replicates).


The following figure supplements are available for figure 5:

**Figure supplement 1.** Fibrillar samples of 15–25 WT and 19–29 S20G do not contain detectable amyloid oligomers.


**Figure supplement 2.** (−)-epigallocatechin gallate (EGCG), a flavanol known to mitigate full-length hIAPP cytotoxicity by preventing it from forming fibrils, likewise mitigates 19–29 S20G cytotoxicity by preventing it from forming fibrils.


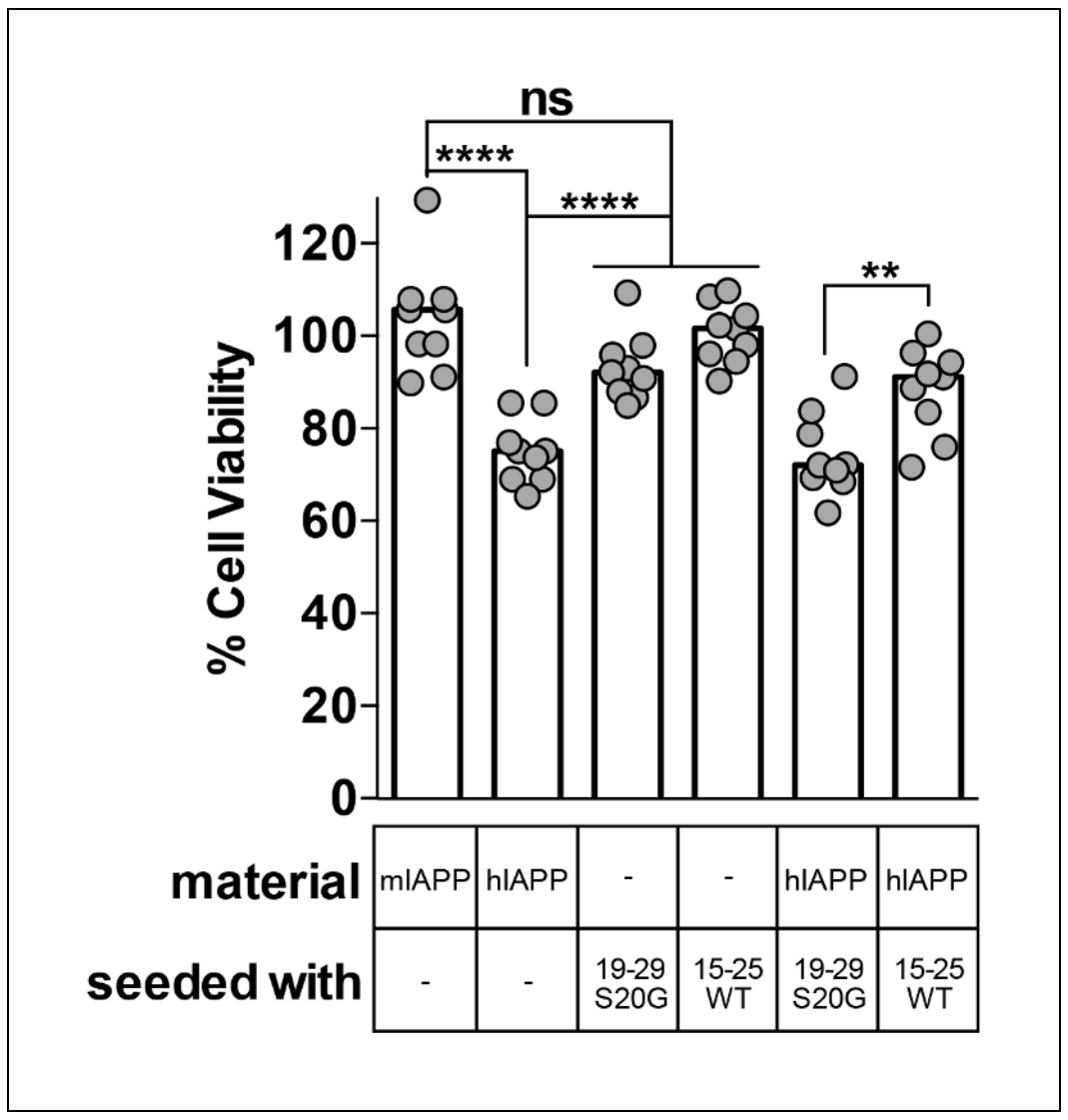


**Figure 6.** Fibril seeds of 15–25 WT reduce the cytotoxicity of full-length hIAPP. In this experiment, we incubated 10 μM hIAPP with or without 10% monomer equivalent of pre-formed seeds overnight under quiescent conditions, the same conditions used to seed full-length hIAPP fibril formation in ***Figure 1***. Next, we diluted the samples 1 to 10 in culture media containing pre-plated Rin5F cells. Note: the concentration of IAPP used in this experiment is less than the IAPP concentrations used in the cytotoxicity assays in ***Figures 1*** and ***5***. hIAPP seeded with stable, toxic 19–29 S20G fibrils is more cytotoxic to Rin5F cells than hIAPP seeded with labile, non-toxic 15–25 WT fibrils. Columns indicate median cell viability (ns = not significant; **p=0.006; ****p<0.0001 using an unpaired t-test with equal standard deviations, n = 9 across three biological replicates, each with three technical replicates). 19–29 S20G seeds and 15–25 WT seeds (100 nM each) are not cytotoxic to Rin5F cells.
The following figure supplement is available for figure 6:

**Figure supplement 1.** hIAPP fibrils made by seeding with each spine segment have slightly different structural features.

are thought to contribute to pancreatic β-cell death during T2D. Fourth, the early onset S20G mutation confers greater cytotoxicity within segment 19–29 and within full-length hIAPP. Last, EGCG, a flavanol that mitigates full-length hIAPP fibril formation and cytotoxicity, likewise mitigates 19–29 S20G fibril formation and cytotoxicity. These results, taken together with the canonical pathogenic amyloid fibril architecture of segment 19–29 S20G, suggest it represents the toxic amyloid spine of hIAPP.

Our studies begin to provide a framework for understanding which hIAPP fibril polymorphs may contribute to pancreatic β-cell death during T2D. Previous structural studies of hIAPP protein segments (***Wiltzius et al., 2008***, ***2009a***; ***Soriaga et al., 2015***) and full-length hIAPP (***Luca et al., 2007***; ***Weirich et al., 2016***; ***Goldsbury et al., 1997***; ***Kajava et al., 2005***; ***Bedrood et al., 2012***; ***Wineman-Fisher et al., 2015***) identified an array of structures with diverse side-chain and sheet arrangements; the 15 hIAPP protein segment structures that overlap the hIAPP amyloid spine belong to six different steric-zipper classes (***Figure 7***). These multiple diverse structures suggest there is significant polymorphism within the hIAPP amyloid spine, but exactly which of these polymorphs elicit cytotoxicity was not known. By studying the structures and cytotoxic effects of protein segments in parallel, we identify a cytotoxic hIAPP fibril structure that may contribute to pancreatic β-cell death during T2D. Additionally, our studies suggest that not all hIAPP fibril structures are cytotoxic.

Both atomic structures presented here reveal a new and potentially important observation: curved β-sheets. In the dry interface of the 19–29 S20G atomic structure, the curved β-sheets accommodate the tightly packed hydrophobic core, which includes a bulky phenylalanine, while maintaining high shape complementarity and large buried surface area. Paradoxically, in the 15–25 WT atomic structure, the curved β-sheets appear to have an opposite effect: the curved β-sheets appear to prevent adjacent sheets from associating to form a canonical pathogenic amyloid fibril architecture. In both atomic structures, the effect of the curved β-sheets is dictated by the registry of adjacent β-sheets (***Supplementary file 1***, ***Figure 7***).

The 15–25 WT atomic structure joins the recently discovered class of out-of-register protein segment structures, which exert disparate cytotoxic effects. Here, we show that 15–25 WT is not cytotoxic but in contrast, the out-of-register protein segment KDWSFY from β2-microglobulin elicits mild cytotoxicity (***Liu et al., 2012***). One notable difference between the two structures is that the 15–25 WT structure is formed of single sheets, while the KDWSFY structure is formed of sheets mated by a dry interface. The dry interface of the KDWSFY atomic structure results in a higher solvation energy per strand compared to the 15–25 WT atomic structure (122 cal/mol/strand vs. 19 cal/mol/strand). Given that cytotoxic structures like 19–29 S20G have relatively high solvation energies per strand

| $_{8}$ATQRLANFLVHSSNNFGAILSSTNVGSNTY$_{37}$<br>**G** | Zipper symmetry class | Parallel (//) Or Anti-Parallel (A//) | Registry |
|---|---|---|---|
| ANFLVH | 2 | // | in |
| NFLVHS | 7 | A// | in |
| NFLVHSS | 6 | A// | in |
| FLVHSSNNFGA | 6 | A// | out |
| LVHSSN | 7 | A// | in |
| HSSNNF | 4 | // | in |
| S**G**NNFGAILSS | 1 | // | in |
| NNFGAIL | 1 | // | in |
| NFGAILS | 7 | A// | out |
| FGAILSS | 6 | A// | in |
| AILSST | 8 | A// | in |
| SSTNVG | 1 | // | in |
| NVGSNTY | 1 | // | in |

**Figure 7.** Schematic of structural features of all hIAPP protein segment structures determined to date. Parallel (//) or Anti-parallel (A//) refers to the orientation of β-strands within β-sheets. Registry refers to the translational offset of β-sheets perpendicular to the fiber axis.

(279 cal/mol/strand; ***Supplementary file 1***), this difference may explain the disparate cytotoxic effects of the two out-of-register structures. However, we need more studies of out-of-register protein structures and their cognate cytotoxic effects to definitively make this conclusion. The disparate cytotoxic effects within this structure class lead us to believe that the nature of cytotoxicity is not simply conferred by in-register or out-of-register structures. As many studies have suggested, there may be more than one mechanism of amyloid-related toxicity and the different mechanisms may be catalyzed by different architectures. Alternatively, maybe if additional residues were included, the anti-parallel out-of-register fiber could be stabilized, thereby increasing its toxicity.

Although the 19–29 WT fibrils prepared in this study appear morphologically similar to 19–29 S20G fibrils, the 19–29 WT fibrils are likely polymorphic and may contain some fraction of fibrils that are structurally similar to non-toxic 15–25 WT fibrils. Previous structural studies of segment 20–29 WT fibrils show that it forms an array of polymorphs, some of which are similar to the 15–25 WT atomic structure (***Griffiths et al., 1995***; ***Jack et al., 2006***; ***Madine et al., 2008***; ***Ashburn et al., 1992***; ***Nielsen et al., 2009***). Structural polymorphism of 19–29 WT fibrils may explain their lower cytotoxicity than 19–29 S20G fibrils, which are homogenous in structure.

These findings, expedited by MicroED, may inform our understanding of hIAPP fibril structures that contribute to pancreatic β-cell death in Type-II Diabetes patients. Going forward, we can use our toxic amyloid spine model as a template for structure-based design in the effort to develop much needed therapeutics that protect against pancreatic β-cell death and disease progression (***Sievers et al., 2011***; ***Kahn et al., 2014***). In addition, if hIAPP fibrils truly are a major type of toxic aggregate that contributes to T2D, then raising antibodies against hIAPP fibrils may represent a

promising strategy for therapeutic development, especially in light of the recent success of preliminary studies with antibodies raised against amyloid-β (***Sevigny et al., 2016***).

## Materials and methods

### IAPP and protein segments

Human IAPP(1–37)-$NH_2$ wild-type and mouse IAPP(1–37)$NH_2$ wild-type were synthesized by Innopep (San Diego, CA) and CS Bio (Menlo Park, CA) and purified to greater than 98% purity. Human and mouse IAPP were prepared by dissolving the lyophilized proteins at 0.25–1 mM in 100% HFIP and leaving them to dissolve for several hours to ensure complete solubility. Next, the HFIP was removed with a CentriVap Concentrator (Labconco, Kansas City, MO). After removal of the HFIP, the peptides were dissolved at 1 mM, 5 mM, or 10 mM in 100% DMSO. The DMSO peptide stocks were diluted 100-fold in filter-sterilized Dulbecco's PBS (Cat. # 14200–075, Life Technologies, Carlsbad, CA). Samples were incubated at room temperature for the designated time periods.

All four spine segments were synthesized by GenScript (Piscataway, NJ) and purified to greater than 98% purity. Fibrils were formed by dissolving lyophilized peptide at 1 mM in PBS and 1% DMSO.

### Crystallization

15-FLVHSSNNFGA-25 (15–25 WT). 15–25 WT was dissolved at 20 mg/ml in ice-cold, nano-pure water and then spin-filtered. Crystals were grown using the hanging drop vapor diffusion method at 4°C in 0.35 M NaSCN and 35% MPD. Crystals grew within several hours and reached maximum size in a week. 3D crystals only a few hundred nanometers thick grew alongside microcrystals in the same drops.

19-SGNNFGAILSS-29 (19–29 S20G). Microcrystals were grown using the hanging drop vapor diffusion method at 30°C in 0.2M acetate salts and 40% MPD. 3D crystals only a few hundred nanometers thick were grown in batch by dissolving lyophilized peptide at 1 mM in PBS and 1% DMSO without seeding. Crystals grew on the bench top at room temperature in several hours.

### MicroED data collection

The procedures for MicroED data collection and processing largely follow published procedures (***Hattne et al., 2015***; ***Shi et al., 2016***). Briefly, a 2–3 µL drop of crystals in suspension was deposited onto a Quantifoil holey-carbon EM grid then blotted and vitrified by plunging into liquid ethane using a Vitrobot Mark IV (FEI, Hillsboro, OR). Blotting times and forces were optimized to keep a desired concentration of crystals on the grid and to avoid damaging the crystals. Frozen grids were then either immediately transferred to liquid nitrogen for storage or placed into a Gatan 626 cryo-holder for imaging. Images and diffraction patterns were collected from crystals using an FEG-equipped FEI Tecnai F20 TEM operating at 200 kV and fitted with a bottom mount TVIPS TemCam-F416 CMOS-based camera. Diffraction patterns were recorded by operating the detector in a movie mode termed 'rolling shutter' with 2×2 pixel binning (***Nannenga et al., 2014b***). Exposure times for these images were either 2 or 3 s per frame. During each exposure, crystals were continuously unidirectionally rotated within the electron beam at a fixed rate of 0.3 degrees per second, corresponding to a fixed angular wedge of 0.6 or 0.9 degrees per frame.

Crystals that appeared visually undistorted and that were 100–300 nm thick produced the best diffraction. Datasets from individual crystals were merged to improve completeness and redundancy. Each crystal dataset spanned a wedge of reciprocal space ranging from 40–80°. We used a selected area aperture of approximately 1 µm. The geometry detailed above equates to an electron dose rate of less than 0.01 $e^{-}/Å^2$ per second being deposited onto our crystals.

Measured diffraction images were converted from TVIPS format into SMV crystallographic format, using in-house software (available for download at http://cryoem.janelia.org/downloads) (***Hattne et al., 2015***).

We used XDS to index and integrate the diffraction images and XSCALE (***Kabsch, 2010***) for merging and scaling together datasets originating from different crystals.

For 19–29 S20G, data from six crystals were merged to assemble the dataset used for molecular replacement. Of note, the resolution was cut off at 1.9 Å to facilitate subsequent rounds of structure refinement.

For, 15–25 WT, data from six crystals were merged to assemble the dataset used for molecular replacement. Of note, the diffraction pattern from the 15-25 WT crystals diffracted with MicroED reveal a pseudo two-fold symmetry. In line with this observation, we indexed and integrated the diffraction images with space group C2, but the datasets had relatively poor statistics compared to the P1 datasets and our attempts at refining molecular replacement solutions from the C2 datasets failed.

## Structure determination

19–29 S20G. We determined the structure using molecular replacement. An idealized 7-residue poly-alanine strand led us to our atomic model. The solution was identified using Phaser (***McCoy, 2007***). A dataset merged from six crystals was used to identify the initial model, but subsequent rounds of model building and refinement were carried out using a dataset from a single crystal. Free R flags were copied over from the dataset merged from six crystals to the single crystal dataset. Subsequent rounds of model building and refinement were carried out using COOT and Phenix, respectively (***Emsley and Cowtan, 2004***; ***McCoy et al., 2005***). Electron scattering factors were used for refinement.

15–25 WT. We determined the structure using molecular replacement. Dozens of search models were used, but an out-of-register β-strand model led us to our solution. The solution was identified using Phaser (***McCoy, 2007***). Subsequent rounds of model building and refinement were carried out using COOT and Phenix, respectively (***Emsley and Cowtan, 2004***; ***McCoy et al., 2005***). Electron scattering factors were used for refinement. To aid in model building, we used a feature enhanced map (FEM), which sharpens B factors at high resolution (***Afonine et al., 2015***).

Calculations of the area buried and shape complementarity (SC) were performed with AREAIMOL (***Lee and Richards, 1971***; ***Collaborative Computational Project, Number 4, 1994***) and SC (***Connolly, 1983***; ***Richards, 1977***; ***Lawrence and Colman, 1993***), respectively.

## ThT binding

30 µL of human and mouse IAPP preparations used in the cytotoxicity assays in ***Figure 1*** were pipetted into a black-wall 384-well plate and then mixed with 3 µL of 1 mM Thioflavin-T (ThT). Fluorescence was recorded with an excitation wavelength of 444 nm and an emission wavelength of 482 nm.

## Dot blot assay

1 µL of each sample generated for cytotoxicity assays in ***Figure 1*** and ***Figure 5A and B*** was applied to a nitrocellulose membrane (Cat. # 162–0146, BioRad, Hercules, CA). Next, the membrane was blocked in 5% (w/v) nonfat dry milk in PBS-T (T = 0.1% (v/v) Tween-20 (Cat. #BP337-500, Fisher)) for 1 hr at room temperature. After blocking, the membrane was incubated with a 1:100 dilution of LOC polyclonal rabbit serum (Pacific Immunology, Ramona, CA) in 5% (w/v) milk in PBS-T at 4˚C overnight. The membrane was washed in PBS-T for 10 min three times, and then incubated with anti-rabbit secondary antibody (RRID:AB_2307391; Cat. #111-035-144, Jackson ImmunoResearch, West Grove, PA) diluted 1:10,000 in PBS-T for 1 hr at RT. The membrane was washed three more times, and then the signal was developed with Clarity Western ECL Substrate (Cat. #170–5061, Bio-Rad) and documented with a CCD camera. Exposures ranging from 5 s to 5 min were collected, but the 5 min exposure was used in all figures.

## Imaging and negative stain transmission electron microscopy

Samples were spotted onto grids (holey or non-holey) and allowed to settle on the grid for 160 to 180 s. Remaining liquid was wicked off and grids were left to dry before analyzing. Sample grids were analyzed on the TF20 Electron Microscope (FEI, Hillsboro, OR). Images were collected at 3500 or 6000x magnification with an additional 1.4x post-column magnification and recorded using a TIETZ F415MP 16 megapixel CCD camera.

Samples for negative-stain EM were spotted on non-holey carbon-coated grids andallowed to settle on the grid for 160 to 180 s. Remaining liquid was wicked off and then 2% uranyl acetate was applied to the grid. After 1 min, the uranyl acetate was wicked off. The grids were left to dry before analyzing on the T12 Electron Microscope (FEI). Images were collected at 3,200 or 15,000x magnification and recorded using a Gatan 2kX2k CCD camera.

## Cell culture

Rin5F cells were purchased from ATCC (RRID:CVCL_2177; Cat. # CRL-2058, Manassas, VA). Cells were cultured in RPMI media (ATCC, Cat. # 30–2001) plus 10% heat-inactivated fetal bovine serum. Cells were cultured at 37°C in a 5% $CO_2$ incubator. They tested negative for mycoplasma using a MycoAlert PLUS Detection Kit (Cat. #: LT07-701, Lonza, Switzerland) and they were authenticated using Cytochrome C Oxidase 1 (COX1) gene analysis by Laragen (Culver City, CA).

HEK293 c18 cells (hereon referred to as HEK293) were a gift from Carol Eng in the laboratory of Arnold J. Berk at UCLA, but they were originally purchased from ATCC (RRID:CVCL_6974). Cells were cultured in DMEM media (Cat. # 11965–092, Life Technologies) plus 10% heat-inactivated fetal bovine serum and 1% pen-strep (Life Technologies). Cells were cultured at 37°C in a 5% $CO_2$ incubator. They tested negative for mycoplasma using a MycoAlert PLUS Detection Kit and they were authenticated using STR profiling (Laragen).

CHO cells were purchased from ATCC (RRID:CVCL_0214; Cat. #: CCL-61). Cells were cultured in RPMI 1640 with 11 mM glucose (Sigma) with 10% FBS, and 1% pen-strep. Cells were cultured at 37°C in a 5% $CO_2$ incubator. They tested negative for mycoplasma using a PCR-based method and they were authenticated using mRNA analysis.

## Spine segment fibril formation

Spine segments were dissolved at 1 mM in PBS with 1% DMSO. Samples were incubated at room temperature for 15 hr or up to one week under quiescent conditions to form fibrils. The presence of fibrils was confirmed with electron microscopy. Fibril samples were diluted appropriately for cell viability assays and fibril formation assays.

## Fiber diffraction and radial profile analysis

Fibrils were spun down and washed with water three times to remove any salt. Fibrils of spine segments were spun down using a tabletop microfuge. Full-length hIAPP fibrils and spine segment seeds were spun down using an Airfuge Ultracentrifuge set at 75,000 rpm for 1 hr (Beckman-Coulter, Brea, CA). The samples were concentrated 10x in water and applied between two capillary ends and then the samples were left to dry overnight. Dried fibrils of spine segments and full-length hIAPP in ***Figure 3D*** were analyzed with a RIGAKU R-AXIS HTC imaging plate detector using Cu K(alpha) radiation from a FRE+ rotating anode generator with VARIMAX HR confocal optics (Rigaku, Tokyo, Japan). Fiber diffraction from full-length hIAPP fibrils used in ***Figure 6*** was recorded by an ADSC Q315 CCD detector at the Advanced Photon Source 24-ID-E beamline (Argonne, IL).

Radial profiles were calculated using a program written in-house. The program calculates the average intensity as a function of distance from the beam center.

## Thioflavin-T assays

Thioflavin-T (ThT) assays were performed in black 96-well plates (Nunc, Rochester, NY) sealed with UV optical tape. hIAPP and mIAPP were dissolved at 1 mM in 100% HFIP. The peptides were then diluted 100-fold in 20 mM sodium acetate pH 6.5 and 10 μM ThT. Unsonicated fibril seeds were added at 1 μM monomer equivalent concentration (10% v/v). ThT fluorescence was recorded with excitation and emission of 444 nm and 482 nm, respectively, using a SpectraMax M5 (Molecular Devices, Sunnyvale, CA). Experiments were performed in quadruplicate and readings were recorded every 3 min.

## Model building and energy analysis of 19–29 WT and 19–29 S20G

To investigate whether 19–29 WT could form a similar structure to 19–29 S20G, we modeled a serine at position 20 in the 19–29 S20G atomic structure. We adjusted the backbone torsion angles so that they fell within the 'allowed' regions of the Ramachandran plot for a non-glycine residue

(*Emsley and Cowtan, 2004*). We performed energy minimization using FoldIt (RRID:SCR_003788) (*Cooper et al., 2010*) and compared the energies of the resulting models of 19–29 WT and 19–29 S20G.

## Cytotoxicity assays

HEK293 cells and Rin5F cells were plated at 10,000 and 27,000 cells per well in 90 µL, respectively, in 96-well plates (Cat. # 3596, Costar, Tewksbury, MA). Cells were allowed to adhere to the plate for 20–24 hr.

For the assay in *Figures 1* and 50 µM full-length IAPP was aged in vitro for the designated incubation times. To generate the soluble and insoluble fractions, the 'hIAPP 24 h' preparation was centrifuged at 21,000xg for 45 min and then the supernatant, which is the soluble fraction, was carefully removed and transferred to a 0.1 µm spin filter tube. Next, the supernatant was filtered and the pelleted material, which is the insoluble fraction, was resuspended in the original total volume.

For the assays in *Figure 5* and *Figure 5—figure supplement 2*, 1 mM spine segment and 100 µM full-length IAPP samples were generated by preparing the samples as described previously and then incubating them for 15 hr at room temperature under quiescent conditions. After the incubation period, the spine segments were diluted appropriately.

For all assays, 10 µL of sample was added to cells. By doing this, samples were diluted 1/10 from in vitro stocks. Experiments were done in triplicate.

The appropriate statistical test for significance was determined by assessing whether (1) The sample sets had a Gaussian distribution using a D'Agostino-Pearson omnibus normality test and (2) The sample sets had equal variance using a Bartlett's test or F test. For samples with Gaussian distributions and equal variances, we employed an unpaired t-test with equal standard deviations. For samples with Gaussian distributions, but unequal variances, we employed an unpaired t-test with Welch's correction. For samples with non-Gaussian distributions and unequal variances, we employed a Mann-Whitney U-test.

## 3-(4,5-dimethylthiazol-2-yl)−2,5-diphenyltetrazolium bromide (MTT) dye reduction assay for cell viability

After a 24 hr incubation of samples with cells, 20 µL of Thiazolyl Blue Tetrazolium Bromide MTT dye (Sigma, St. Louis, MO) was added to each well and incubated for 3.5 h at 37°C under sterile conditions. The MTT dye stock is 5 mg/mL in Dulbecco's PBS. Next, the plate was removed from the incubator and 100 µL of MTT stop solution (Cat. #4101, Promega, Madison, WI) was added to each well. We ensured the MTT crystals were fully dissolved by placing the plates on an orbital shaker (slow speed) for about an hour prior to taking measurements. Alternatively, the MTT assay was stopped by carefully aspirating off the culture media and adding 100 µL of 100% DMSO to each well. Absorbance was measured at 570 nm using a SpectraMax M5. A background reading was recorded at 700 nm and subsequently subtracted from the 570 nm value.

Cells treated with vehicle alone (PBS + 0.1% DMSO) were designated at 100% viable, and cell viability of all other treatments was calculated accordingly.

For the MTT reduction assay in *Figure 6*, a single data point from the mIAPP sample set was deemed an outlier based on 2 lines of evidence: (1) The data point was identified as an outlier using a Grubb's test ($\alpha$ = 0.1) for outliers using the n = 9 sample set, and (2) When the sample set was pooled with more data collected for different experiments (n = 42), the data point was identified as an outlier using a more stringent Grubb's test ($\alpha$ = 0.01).

## Caspase-3/7 activation assay

We used the caspase-3/7 GLO assay (Cat. # G8091, Promega, Madison, WI) to detect caspase-3/7 activation. For this assay, Rin5F cells were plated as previously described in white-walled 96-well plates (Cat. # 3917, Costar, Tewksbury, MA). After the designated aging period of each hIAPP preparation, 10 µL of sample was added to cells and thus diluted 1/10 from in vitro stocks. Experiments were performed in triplicate. Samples were incubated with cells for 24 h. Next, cell culture media, caspase-3/7 reagent, and the cells were brought to room temperature. All media was aspirated from wells and then replaced with 25 µL of media and 25 µL of caspase-3/7 reagent and mixed thoroughly. The plate was incubated at room temperature for 30 min and then luminescence was

measured using a SpectraMax M5. Experimental points were normalized to vehicle-treated cells, which were designated as 100%. Cells treated with 2 μM staurosporine were used as a positive control to ensure the assay kit worked correctly.

## FRET-based real-time monitoring of caspase-3 activity

CHO cells were stably transfected with a vector producing EYFP and ECFP connected via a short linker containing the Asp-Glu-Val-Asp (DEVD) sequence targeted by activated caspase-3. The short linker allows fluorescence energy transfer (FRET) to occur between the two fluorophores. During apoptosis activated caspase-3 cleaves the linker resulting in a loss of FRET measured as a reduced 540 nm/480 nm emission ratio.

Cells were plated at 25,000 cells per well in black 96-well optical bottom plates (Nunc, Grand Island, NY) and the assay was performed in Krebs-Ringer (120 mM NaCl, 4.7 mM KCl, 2.5 mM $CaCl_2$, 1.2 mM $MgSO_4$, 0.5 mM $KH_2PO_4$, pH 7.4) supplemented with 20 mM HEPES and 2 mM glucose (KRHG).

hIAPP peptides (1–37, 15–25 WT and S20G, 19–29 WT and S20G) (final peptide concentration 50 μM in 1% DMSO) were mixed with sonicated, preformed fibrils (seeds) made of the same peptide (corresponding to 166 nM of monomers) and immediately added to the plated cells. FRET was monitored in real-time by measuring emission at 480 nm and 540 nm with 440 nm excitation in a FLUOstar Omega microplate reader (BMG Labtech) over 24 hr at 37°C.

## SDS sensitivity assay

Fibrils of 15–25 WT and 19–29 S20G at monomer equivalent concentrations were allowed to form for one week to ensure complete fibril formation. The samples were homogenized with vortexing, and then aliquoted to 0.5 mL tubes with equal volumes. Each fibril sample was treated with water or increasing amounts of SDS, and then heated at 55°C for 20 min. Next, an aliquot of each sample was transferred to a 384-well plate and turbidity was measured by recording absorbance at 340 nm. Each fibril sample was spotted onto a grid for negative-stain EM to analyze fibril abundance. The experiment was repeated twice, but the results of 1 experiment are shown in ***Figure 4—figure supplement 1***.

# Acknowledgements

We thank Dan Anderson for suggesting the use of FEM maps. We thank the Advanced Photon Source (APS) staff for beamline support during collection of fiber diffraction: M. Capel, K. Rajashankar, N. Sukumar, J. Schuermann, I. Kourinov and F. Murphy at NECAT beam lines 24-ID; the UCLA-DOE X-ray Crystallography Core Technology Center, Michael Collazo, and the UCLA-DOE Macromolecular Crystallization Core Technology Center for setting up initial crystallization screens; Ivo Atanasov and the Electron Imaging Center for NanoMachines (EICN) of California NanoSystems Institute (CNSI) at UCLA for the use of their electron microscopes; and the UCLA Statistical Consulting group for guidance on statistical computing. The UCLA-DOE X-ray and Macromolecular and Macromolecular Crystallization Core Technology Centers are supported in part by the Department of Energy grant DE-FC0302ER63421. Use of the Advanced Photon Source is supported by National Institutes of Health grants P41 RR015301 and P41 GM103403 and Department of Energy under Contract DE-AC02-06CH11357. We thank the Cure Alzheimer's Fund, the Swedish Research Council (VR 2015–02297), the Swedish Diabetes Foundation, the Janelia Research Campus visitor program, HHMI, and the National Institutes of Health (NIH) (R01 AG029430) for support.

# Additional information

## Funding



| Funder | Grant reference number | Author |
|---|---|---|
| National Institutes of Health | R01 AG029430 | Pascal Krotee<br>Jose A Rodriguez<br>Michael R Sawaya<br>Duilio Cascio |

| | | |
|---|---|---|
| | | Sarah Griner<br>David S Eisenberg |
| Howard Hughes Medical Institute | | Pascal Krotee<br>Jose A Rodriguez<br>Michael R Sawaya<br>Duilio Cascio<br>Francis E Reyes<br>Dan Shi<br>Johan Hattne<br>Brent L Nannenga<br>Sarah Griner<br>Tamir Gonen<br>David S Eisenberg |
| Vetenskapsrådet | VR 2015-02297 | Marie E Oskarsson<br>Gunilla T Westermark |
| Swedish Diabetes Foundation | | Marie E Oskarsson<br>Gunilla T Westermark |
| Cure Alzheimer's Fund | | Stephan Philipp<br>Charles G Glabe |

The funders had no role in study design, data collection and interpretation, or the decision to submit the work for publication.


### Author contributions

PK, Conception and design, Acquisition of data, Analysis and interpretation of data, Drafting or revising the article, Contributed unpublished essential data or reagents; JAR, CGG, Acquisition of data, Analysis and interpretation of data, Contributed unpublished essential data or reagents; MRS, SP, Acquisition of data, Analysis and interpretation of data, Drafting or revising the article, Contributed unpublished essential data or reagents; DC, MEO, TG, Acquisition of data, Analysis and interpretation of data; FER, DS, JH, BLN, Acquired MicroED diffraction data, Acquisition of data; SG, Performed preliminary, unpublished dot blot assays for initial cytotoxicity studies; LJ, Conception and design, Analysis and interpretation of data; GTW, Suggested the FRET-based bionsensor assay; Analyzed and interpreted FRET-based biosensor assay results; Helped compose responses to reviewers' comments, Analysis and interpretation of data; DSE, Conception and design, Analysis and interpretation of data, Drafting or revising the article

### Author ORCIDs


Pascal Krotee, http://orcid.org/0000-0002-9582-658X
Michael R Sawaya, http://orcid.org/0000-0003-0874-9043
Duilio Cascio, http://orcid.org/0000-0002-3877-6803
Johan Hattne, http://orcid.org/0000-0002-8936-0912


## Additional files

### Supplementary files

• Supplementary file 1. Summary of structural features and biophysical properties of all hIAPP protein segment structures determined to date.

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
