## [Decision Letter]

Thank you for submitting your article "Atomic Structures of Fibrillar Segments of Islet Amyloid Polypeptide Suggest Tightly Mated β-Sheets Drive Cytotoxicity" for consideration by *eLife*. Your article has been reviewed by three peer reviewers, and the evaluation has been overseen by a Reviewing Editor, Nir Ben-Tal, and Michael Marletta as the Senior Editor. The reviewers have opted to remain anonymous.

The reviewers have discussed the reviews with one another and the Reviewing Editor has drafted this decision to help you prepare a revised submission.

Summary:

The manuscript addresses the important and controversial topic of human islet amyloid polypeptide (hIAPP) toxicity via a mostly structural study. The work is technically excellent, the use of Cryo electron microscopy-based MicroED is highly suitable for the scientific quest and the addition of the biological data is very important. The main conclusion is that the fibrillar form of hIAPP segment 19-29S20G is toxic, but its oligomeric form is not. In addition, in- or out-of registration/hydrophobicity of the four IAPP segments of interest in fibrillation is insufficient to account for the observed differential toxicity. This study concludes that 19-29S20G could be used as a model system for examining hIAPP aggregation and toxicity. The work is elegant and it provides important insights on the cytotoxicity of the polypeptide that is associated with Type II diabetes. The conclusions are consistent with the experimental results and the Discussion is very good. On the other hand, it is somewhat surprising that this manuscript heavily emphasizes on cytotoxicity – a not so very original point of this study, while the more novel structural observations are only mentioned in passing. If revised, it could be suitable for *eLife*.

Essential revisions:

1) The authors make it appear as if fibrils (and not oligomers or other intermediates) are generally considered as the major cytotoxic species of IAPP. However, there is very abundant literature evidence that oligomers and not fibrils are the major cytotoxic species, and the current study also does not explicitly analyze oligomer samples. The current description might be motivated by bad experiences with journal editors or referees who frequently dismiss all work on fibrils as being irrelevant and notoriously believe the pathology of any protein misfolding disease to solely depend on oligomers. This concept, while being too simple minded, is wide spread and causes numerous problems when attempting to publish solid scientific work. Yet, I also think that it is problematic to neglect the considerable body of published evidence in favor of the toxicity of IAPP oligomers, although there are also studies reporting the in vitro toxicity of fibrils, at least in certain set ups. Therefore, the following sentences should be changed:

"Researchers have accumulated substantial evidence that supports a causative link between hIAPP aggregation and islet cell death leading to T2D" and "Consistent with this causation, mouse IAPP does not aggregate and mice do not get T2D. However, mice can be induced to develop islet amyloid and T2D when they are engineered to express human IAPP and fed a high fat diet (Verchere et al., 2996; Westermark et al., 2000). Perhaps the strongest support for a causative link is the mutation in hIAPP, hIAPP-S20G; segments containing this mutation aggregate more quickly and lead to early onset T2D in families who carry this lesion" suggest that IAPP toxicity causes T2D (type 2 diabetes). We think that no medical scientist would agree to this notion. Instead, IAPP aggregation is rather thought to constitute a secondary complication of T2D that actually develops for other reasons. The above sentences should be removed or changed such that they no longer imply that IAPP aggregation causes T2D.

"The causative link between hIAPP amyloid fibril formation and islet cell death is widely accepted, but researchers have debated whether soluble, pre-fibrillar oligomers or amyloid fibrils are the primary cytotoxic species that drive islet cell death" and "researchers suggested that amyloid fibrils are the primary cytotoxic species because hIAPP is only cytotoxic in its fibrillar form", "if hIAPP fibrils are the primary cytotoxic species, then determining the atomic structure of the spine of hIAPP fibrils is a logical approach for advancing our understanding of disease-relevant targets", and "we focused on studying fibrillar structures of hIAPP as the more relevant to cytotoxicity". These statements turn the currently dominating belief in the field rather upside down. Pre-fibrillar oligomers and other intermediates (and not fibrils) are nowadays considered to be the major toxic species of IAPP, although this evidence is largely based on in vitro studies and the in vivo pathogenic agents remain to be established. This view can be found, for example, in the very influential review from G. Westermark (Physiological Reviews 2011 Vol. 91 no. 3, 795-826), one of the coauthors of the current study, where it says that "the mature amyloid fibril is presumed to be relatively inert and to have no significant cell toxicity. Rather, smaller oligomeric intermediates formed during fibrillogenesis are thought to be cytotoxic". The above statements should be amended.

The following statement was used to support that residues 15-29 are important for toxicity: "previous work by our laboratory has shown that Phe15 is required for stabilizing an on-pathway α-helical dimer and that mutating this residue can delay aggregation". However, it solely describes aggregation and not toxicity. Please amend.

"The cytotoxicity of hIAPP resides mainly in its fibrillar form" and "here, we offer support for mature amyloid fibrils as a major type of aggregate responsible for islet cell death by showing that fibrillar hIAPP is significantly more cytotoxic to cultured pancreatic β-cells than pre-fibrillar forms". These statements are questionable as the presently reported cytotoxicity measurements did not explicitly test oligomer samples.

The origin of divergence in the literature regarding the identity of the toxic entity perhaps has to do with the rapid fibrillation rate of hIAPP compared with, say, amyloid-β, where isolation of hIAPP monomers, oligomers, protofibrils and mature fibrils is nontrivial, as the authors also acknowledged. Since toxicity is the biological endpoint of a chemical (e.g., ROS), physical (amyloids, oligomers and their associated hydrophobicity/membrane partitioning, assembly kinetics, assembly with intra- and extracellular proteins) or biological (the triangle between insulin, amylin and glucose etc.) trigger, it seems to make more sense that it is the combined contributions of these aspects, different for hIAPP fibrils and oligomers in each aspect, induce the toxicity of hIAPP.

2) If indeed the fibrillar form of the polypeptide is the toxic element, could antibodies raised against this form be a treatment for Type II diabetes? Unlike the case of amyloid-β vaccination, the risk of meningoencephalitis is not a concern.

3) What could be the cause that out of registration amyloid proteins behave so differently, as acknowledged by the authors in the Discussion (fifth paragraph)? Does it suggest that the structural properties of protein monomers/amyloids may not be the only source to explain toxicity?

4) It might be beneficial to compare the overall hydrophobicity and charge of the four segments to decipher their differential toxicity as reported in this paper.

5) All "cytotoxicity" data in Figure 1 are solely based on the MTT assay, which is known to sometimes produce false positive results. See, for example J Biol Chem. 2014 289: 35781-94. These measurements need to be confirmed with another assay.

6) Figure 1: the "dot blot" with hIAPP is not of suitable quality for evaluation, as it only shows a black image. No conclusion is possible based on these data.

7) Figure 6 is not able to show a templating of toxicity by a fragment onto full-length IAPP. The 10% 19-29 S20G fibril seeds produced the same toxicity as full-length hIAPP without seeds, while 15-25 WT seeds reduced toxicity. The most straightforward interpretation of these findings is probably that 19-29 S20G fibril seeds did not affect the formation of toxic species from hIAPP, while 15-25 WT seeds interact with hIAPP to reduce its toxicity. We do not see how this experiment justifies the conclusion that there is seeding of toxic activity.

8) What could/should be emphasized much more clearly (specifically in the Discussion) are the following issues:

A) The current cross-β sheets are concave or convex when you look down the spine axis. This curvature is a major difference to Madine et al., 2008, and probably also most previous steric zippers consisting of short chains. What does this curvature mean or what enforces it?

B) Associated with this observation, Figure 3 shows different backbone distances at the outer ends of the zipper and in the center. This variation of the packing distance could be an important new observation.

C) Most steric zippers have the strands oriented perpendicular to the sheet long axis. This is not the case in Figure 4. Why?

D) How does the current packing of the zippers relate to the ones seen in earlier zipper described for IAPP (Nature Structural & Molecular Biology 16, 973 – 978, 2009)? Do the small zippers show the same interactions as the large ones? If not, what does it mean for the specificity of the side-chain interactions in the fibril?

---

## [Author Response]

*Essential revisions:*

*1) The authors make it appear as if fibrils (and not oligomers or other intermediates) are generally considered as the major cytotoxic species of IAPP. However, there is very abundant literature evidence that oligomers and not fibrils are the major cytotoxic species, and the current study also does not explicitly analyze oligomer samples. The current description might be motivated by bad experiences with journal editors or referees who frequently dismiss all work on fibrils as being irrelevant and notoriously believe the pathology of any protein misfolding disease to solely depend on oligomers. This concept, while being too simple minded, is wide spread and causes numerous problems when attempting to publish solid scientific work. Yet, I also think that it is problematic to neglect the considerable body of published evidence in favor of the toxicity of IAPP oligomers, although there are also studies reporting the in vitro toxicity of fibrils, at least in certain set ups. Therefore, the following sentences should be changed:*

We have amended the manuscript to note that several current studies of hIAPP have implicated oligomers as the toxic agents (Introduction, third paragraph). Thank you for suggesting this point.

*"Researchers have accumulated substantial evidence that supports a causative link between hIAPP aggregation and islet cell death leading to T2D" and "Consistent with this causation, mouse IAPP does not aggregate and mice do not get T2D. However, mice can be induced to develop islet amyloid and T2D when they are engineered to express human IAPP and fed a high fat diet (Verchere et al., 2996; Westermark et al., 2000). Perhaps the strongest support for a causative link is the mutation in hIAPP, hIAPP-S20G; segments containing this mutation aggregate more quickly and lead to early onset T2D in families who carry this lesion" suggest that IAPP toxicity causes T2D (type 2 diabetes). We think that no medical scientist would agree to this notion. Instead, IAPP aggregation is rather thought to constitute a secondary complication of T2D that actually develops for other reasons. The above sentences should be removed or changed such that they no longer imply that IAPP aggregation causes T2D.*

We have amended the manuscript to remove any assertion of causation. We do, however, point out the evidence showing a correlation of hIAPP aggregation to pancreatic β-cell death (Introduction, second paragraph).

"The causative link between hIAPP amyloid fibril formation and islet cell death is widely accepted, but researchers have debated whether soluble, pre-fibrillar oligomers or amyloid fibrils are the primary cytotoxic species that drive islet cell death".

In the amended manuscript we have removed the term “causative”.

*And "researchers suggested that amyloid fibrils are the primary cytotoxic species because hIAPP is only cytotoxic in its fibrillar form", "if hIAPP fibrils are the primary cytotoxic species, then determining the atomic structure of the spine of hIAPP fibrils is a logical approach for advancing our understanding of disease-relevant targets", and "we focused on studying fibrillar structures of hIAPP as the more relevant to cytotoxicity".*

These sentences have been amended in the revised manuscript (Introduction, third paragraph and subsection “hIAPP preparations that contain fibrils are cytotoxic to cultured rat pancreatic β-cells”, last paragraph.

*These statements turn the currently dominating belief in the field rather upside down. Pre-fibrillar oligomers and other intermediates (and not fibrils) are nowadays considered to be the major toxic species of IAPP, although this evidence is largely based on in vitro studies and the in vivo pathogenic agents remain to be established. This view can be found, for example, in the very influential review from G. Westermark (Physiological Reviews 2011 Vol. 91 no. 3, 795-826), one of the coauthors of the current study, where it says that "the mature amyloid fibril is presumed to be relatively inert and to have no significant cell toxicity. Rather, smaller oligomeric intermediates formed during fibrillogenesis are thought to be cytotoxic". The above statements should be amended.*

Thank you for noting the important difference between in vitro and in vivo studies, which we now mention in the third paragraph of the Introduction. We do not, however, get into the mechanism of hIAPP action on cells; we restrict ourselves to reporting our observations of toxicity without proposing a mechanism. Our private hypothesis is that the mechanism may involve both mature fibrils and smaller species, perhaps short fibrils. That is, amyloid fibril deposition leads to distortion of the histological structure of the affected organ. Islet hormone release is dependent on the islet architecture and regulated insulin release requires cell-cell interaction via connexin-36. When hIAPP amyloid deposits extracellularly it results in separation of the β cells and loss of synchronized hormone release. Islet amyloid occurs perivascular and sometimes prominent deposits occur, possibly influencing diffusion of oxygen and nutrients. Whereas mature amyloid fibrils may exert no direct β cell cytotoxicity, amyloid fibrils somehow affect islet function, and the results of our paper may contribute to the ultimate understanding of the mechanism.

Although we do not propose a mechanism, we report one experiment that may bear on it. With our FRET-based amyloid toxicity assay, we showed that the amyloid fibril formation-process induced toxicity (Figure 5). An earlier study using the same assay revealed that the fibril formation-process is cytotoxic when it took place in contact with the cell surface, and the toxicity ceased when fibril formation was halted (Oskarsson, 2015, JBC). Thus, our data support that hIAPP-amyloid fibril formation induces cell toxicity, but amyloid fibrils neutralize the toxicity.

*The following statement was used to support that residues 15-29 are important for toxicity: "previous work by our laboratory has shown that Phe15 is required for stabilizing an on-pathway α-helical dimer and that mutating this residue can delay aggregation". However, it solely describes aggregation and not toxicity. Please amend.*

We amended the wording to make our intended statement clearer. It now reads: “Third, previous work by our laboratory has shown that Phe15 may be part of the amyloid spine because it is required for stabilizing an on-pathway α-helical dimer and mutating this residue can delay fibril formation (Wiltzius et al., 2009).”

*"The cytotoxicity of hIAPP resides mainly in its fibrillar form" and "here, we offer support for mature amyloid fibrils as a major type of aggregate responsible for islet cell death by showing that fibrillar hIAPP is significantly more cytotoxic to cultured pancreatic β-cells than pre-fibrillar forms". These statements are questionable as the presently reported cytotoxicity measurements did not explicitly test oligomer samples.*

In order to address this revision and essential revisions #5 and #6, we repeated the cytotoxicity studies shown in Figure 1 and Figure 1—figure supplement 1 with some modifications. The most important modification was that we used half the concentration of IAPP that we used in our original studies. This modification allowed us to test the cytotoxicity of oligomer and fibril preparations.

Using the LOC antibody, we detect fibrillar oligomers in the “hIAPP 0 h” preparation. Although the LOC antibody was raised against hIAPP fibrils, previous studies show that it binds to fibrillar oligomers of amyloid-β (Wu, 2010, JBC). Fibrillar oligomers are structurally distinct from pre-fibrillar oligomers, which are recognized by the A11 antibody.

Using two metrics of cytotoxicity, we find that hIAPP preparations that contain these fibrillar oligomers are significantly less cytotoxic than hIAPP preparations that contain fibrils and no detectable oligomers. The full description of these results starts in the subsection “hIAPP preparations that contain fibrils are cytotoxic to cultured rat pancreatic β-cells”.

The above statements were amended to reflect our updated cytotoxicity studies. They now read: “We observe that hIAPP preparations that contain fibrils are significantly more cytotoxic to rat pancreatic β-cells than hIAPP preparations that contain oligomers but no detectable fibrils (Figure 1)”; and “In closer agreement with earlier studies, we find that hIAPP preparations that contain fibrils are cytotoxic to rat pancreatic β-cells, thus motivating us to determine the structure of the spine of hIAPP fibrils.”

*The origin of divergence in the literature regarding the identity of the toxic entity perhaps has to do with the rapid fibrillation rate of hIAPP compared with, say, amyloid-β, where isolation of hIAPP monomers, oligomers, protofibrils and mature fibrils is nontrivial, as the authors also acknowledged. Since toxicity is the biological endpoint of a chemical (e.g., ROS), physical (amyloids, oligomers and their associated hydrophobicity/membrane partitioning, assembly kinetics, assembly with intra- and extracellular proteins) or biological (the triangle between insulin, amylin and glucose etc.) trigger, it seems to make more sense that it is the combined contributions of these aspects, different for hIAPP fibrils and oligomers in each aspect, induce the toxicity of hIAPP.*

Good point. But we do not have evidence for mechanism, so we prefer not to discuss this thoroughly. However, we acknowledge the potential disconnect between toxic aggregates identified in vitro and toxic aggregates present in vivo in the third paragraph of the Introduction and we briefly mention the possibility of multi-variable toxicity in the sixth paragraph of the Discussion.

*2) If indeed the fibrillar form of the polypeptide is the toxic element, could antibodies raised against this form be a treatment for Type II diabetes? Unlike the case of amyloid-β vaccination, the risk of meningoencephalitis is not a concern.*

Presumably, yes. We mention this in the Discussion of this revised submission (last paragraph).

*3) What could be the cause that out of registration amyloid proteins behave so differently, as acknowledged by the authors in the Discussion (fifth paragraph)? Does it suggest that the structural properties of protein monomers/amyloids may not be the only source to explain toxicity?*

One potential cause for the different cytotoxic effects of the 2 out-of-register protein segments is whether or not the out-of-register sheets are mated. Mildly cytotoxic protein segment KDWSFY from β-2 microglobulin has sheets mated by a dry interface, but non-toxic hIAPP 15-25 WT is composed of single sheets. The dry interface of the KDWSFY crystal structure results in higher solvation energy per strand compared to the 15-25 WT crystal structure (122 cal/mol/strand vs. 19 cal/mol/strand). Given that cytotoxic structures like 19-29 S20G have high solvation energies per strand (279 cal/mol/strand; [Supplementary-material SD1-data]), this difference may explain the disparate cytotoxic effects of the two out-of-register structures. However, we need more studies of out-of-register protein structures and their cognate cytotoxicity effects to definitively make this conclusion.

We added this observation to the Discussion of this revised submission (sixth paragraph).

*4) It might be beneficial to compare the overall hydrophobicity and charge of the four segments to decipher their differential toxicity as reported in this paper.*

We thank the reviewers for making this suggestion. To investigate this and to compare the structures of shorter hIAPP protein segments previously determined, we added [Supplementary-material SD1-data], which lists many structural and biophysical features of the 15 protein segment crystal structures from hIAPP, including the structures described in this study.

Spine segments 19-29 S20G and 15-25 WT possess similar charges (0 and 0.1 at pH 7, respectively) and only slightly different hydrophobicity values (0.25 and 0.29, respectively) compared to other hIAPP protein segments for which we have crystal structures. Furthermore, the overall charge and hydrophobicity values for cytotoxic full-length hIAPP, (2.1 at pH 7 and – 0.097) are not particularly similar to any of the charges or hydrophobicity values for the protein segments described here. It is difficult to associate 1 biophysical characteristic with cytotoxicity as we only have cytotoxicity data for the 4 protein segments described in this study. Nonetheless, since we observe that the hydrophobicity values for the protein segments described here are close to the average hydrophobicity value for all hIAPP protein segments ever characterized (average = 0.11), we presume overall hydrophobicity and charge of the segments do not fully explain their differential toxicity. It seems more likely that their differential cytotoxicity is better explained by their divergent structures.

*5) All "cytotoxicity" data in Figure 1 are solely based on the MTT assay, which is known to sometimes produce false positive results. See, for example J Biol Chem. 2014 289: 35781-94. These measurements need to be confirmed with another assay.*

We acknowledge the potential issues of the MTT assay. Of note, in the JBC paper cited by the reviewers, the researchers discovered that only fragmented fibrils, not unfragmented fibrils like those used in our studies, produced false positive results in the MTT assay. Additionally, as the reviewers point out in essential revision #1, our studies described in the original Figure 1 and Figure 1—figure supplement 1 did not explicitly test oligomer samples. Thus, we repeated the cytotoxicity studies with some modifications in order to address these 2 essential revisions and essential revision #6.

By halving the concentration of IAPP that we used in our original studies, we were able to generate oligomer and fibril preparations of hIAPP. In the “hIAPP 0 h” preparation, we detect only fibrillar oligomers and in the “hIAPP 24 h” preparation, we detect only amyloid fibrils.

We tested the cytotoxicity of the samples using the MTT assay and a luminescence-based caspase-3/7 detection assay in parallel. Active caspase-3 and caspase-7 proteases are markers of apoptosis (Budihardjo, 1999, Annu Rev Cell Dev Biol). Both assays show that “hIAPP 24 h,” which contains fibrils and no detectable oligomers, is significantly more cytotoxic than “hIAPP 0 h,” which contains only oligomers. Furthermore, we find that the insoluble fraction of the cytotoxic “hIAPP 24 h” sample is more cytotoxic than the soluble fraction of the sample, providing more evidence for cytotoxicity of fibrils in our studies. We added the results of these studies to Figure 1 and Figure 1—figure supplement 1 in the revised manuscript. A full description of this study is in the subsection “hIAPP preparations that contain fibrils are cytotoxic to cultured rat pancreatic β-cells”.

*6) Figure 1: the "dot blot" with hIAPP is not of suitable quality for evaluation, as it only shows a black image. No conclusion is possible based on these data.*

As stated in our response to essential revisions #1 and #5, we repeated the cytotoxicity studies described in the original Figure 1 and Figure 1—figure supplement 1 using a modified setup in order to address the reviewers’ comments on the MTT assay and the absence of oligomers.

Additionally, we performed dot blot assays of the samples used in the cytotoxicity studies with 25 different conformational antibodies known to recognize soluble oligomers. Only one of these antibodies, LOC, gave a robust and consistent result across biological triplicates. We replaced the original dot blot results with the LOC dot blot results in Figure 1 and Figure 1—figure supplement 1. Professor Charles Glabe and Dr. Stephan Philipp from UC Irvine tested our samples with the 25 different conformational antibodies and thus are now co-authors of the manuscript.

*7) Figure 6 is not able to show a templating of toxicity by a fragment onto full-length IAPP. The 10% 19-29 S20G fibril seeds produced the same toxicity as full-length hIAPP without seeds, while 15-25 WT seeds reduced toxicity. The most straightforward interpretation of these findings is probably that 19-29 S20G fibril seeds did not affect the formation of toxic species from hIAPP, while 15-25 WT seeds interact with hIAPP to reduce its toxicity. We do not see how this experiment justifies the conclusion that there is seeding of toxic activity.*

We agree with the reviewers’ interpretation of the result with hIAPP seeded with toxic 19-29 S20G fibrils. This result does not appear to show a templating of toxicity by a fragment onto full- length hIAPP. However, the result with hIAPP seeded with non-toxic 15-25 WT fibrils may be explained by (1) our interpretation of templating growth of fibrils of relatively low toxicity or (2) by the reviewers’ interpretation of seed interacting with hIAPP to reduce its toxicity. Both interpretations seem reasonable considering that 15-25 WT fibrils template full-length hIAPP fibril formation (Figure 1 in the revised manuscript). We include both interpretations in the revised manuscript. See subsection “Fibril seeds of 15-25 WT reduce the cytotoxicity of full-length hIAPP”, third paragraph.

*8) What could/should be emphasized much more clearly (specifically in the Discussion) are the following issues:*

*A) The current cross-β sheets are concave or convex when you look down the spine axis. This curvature is a major difference to Madine et al., 2008, and probably also most previous steric zippers consisting of short chains. What does this curvature mean or what enforces it?*

This subject is addressed more fully in the amended manuscript. The β-sheets in the crystal structures presented here are curved compared to shorter hIAPP protein segments previously determined. To quantify β-sheet deviation from planarity across all protein segment crystal structures, we calculated their standard deviation of atoms from a best-fit plane ([Supplementary-material SD1-data]). Many of the previous steric-zippers are very flat, but some including the 15-25 WT and 19-29 S20G crystal structures are in the top half of deviation magnitude (Figure 3—figure supplement 2; Figure 4—figure supplement 2). Some of the sheets have sharp kinks rather than gradual curves. 19-29 S20G has both. The meaning of the deviations from planarity is not yet clear. These values are included in the newly added [Supplementary-material SD1-data] and discussed in the Results section.

Our private hypothesis is that kinking and curvature are two mechanisms by which β-sheets may increase the interactions with neighboring sheets. In other words, kinks and curvature impart plasticity to the otherwise strictly planar β-sheet so it may conform to the shape of an adjacent surface to produce tighter dry interfaces (as in the case of 19-29 S20G) or optimize crystal contacts (as in the case of 15-25 WT). Indeed, our modeling studies showed without the kink introduced by the S20G mutation, the 19-29 segment cannot form as tight of a dry interface (see last paragraph in the subsection “Segment 19-29 S20G forms pairs of β-sheets tightly mated by a dry interface”).

We speculate that kinking and curvature are likely to be observed more frequently with increasingly longer segments, since longer segments are likely to have greater variation in side chain lengths and kinks and curves provide additional degrees of freedom for sheets of varying thickness to mate more tightly.

Moreover, we think that kinked sheets are likely to be more common for in-register sheets than out-of-register sheets. We speculate that if the 15-25 WT segment could kink in addition to curving, the new shape would afford another chance that it may conform tightly with neighboring sheets. However, there is likely to be a high energy barrier to introducing a kink in the sheet formed by the 15-25 WT segment because the sheet lacks alignment of glycine residues, which assists with kink formation. Recall that kink formation requires a crease to propagate through the entire length of the sheet. Because the 15-25 WT structure is out of register, Gly24 residues do not align in a continuous line.

*B) Associated with this observation, Figure 3 shows different backbone distances at the outer ends of the zipper and in the center. This variation of the packing distance could be an important new observation.*

Thank you for this suggestion. To investigate this question, we measured the standard deviation in sheet-to-sheet distance for all hIAPP protein segment structures determined to date. The variation of packing distance appears to accommodate the variation in side-chain sizes protruding from the sheet. This observation is noted in the revised text and it is also included in the newly added [Supplementary-material SD1-data].

*C) Most steric zippers have the strands oriented perpendicular to the sheet long axis. This is not the case in Figure 4. Why?*

Indeed, most steric-zippers are formed of β-strands stacked perpendicular to the sheet long axis. We describe such zippers as “in-register.” A few years ago, our group discovered steric-zipper structures that are formed of β-strands that stack at an angle to the sheet long axis. We describe these zippers as “out-of-register.” Deviation of strands from the fibril perpendicular is a natural consequence of the registration shift implied by out-of-register structures. The 15-25 WT crystal structure presented in Figure 4 is an example of the out-of-register-steric-zippers.

To date, only 4 out-of-register steric-zippers have been identified and the significance of the out- of-register stacking of β-strands is not yet well established. It is possible that the out-of-register stacking could infer tendencies for the sheet to roll into a cylinder, like cylindrin does (Laganowsky, 2012, Science). The 4 out-of-register steric-zippers are formed of protein segments derived from β-2 microglobulin (Liu, 2012, PNAS), prion protein (Yu, 2015, Biochemistry), and IAPP (Soriaga, 2015, J Phys Chem Bio) suggesting out-of-register stacking is a shared feature of amyloid proteins.

We expanded our description of out-of-register protein structures in the Results and Discussion sections.

*D) How does the current packing of the zippers relate to the ones seen in earlier zipper described for IAPP (Nature Structural & Molecular Biology 16, 973 – 978, 2009)? Do the small zippers show the same interactions as the large ones? If not, what does it mean for the specificity of the side-chain interactions in the fibril?*

Crystal structures of shorter hIAPP protein segments exhibit an array of sheet and side-chain interactions, which we term polymorphism. Even shorter protein segments that fully overlap with the 11-residue spine segments described in this work exhibit significant polymorphism. We list the steric-zipper classes and other structural features of each protein segment in the newly added Figure 7 and [Supplementary-material SD1-data] and we examine the polymorphism across the structures starting in the fourth paragraph of the Discussion.

Furthermore, full-length hIAPP fibril models derived from solid-state NMR studies (Luca, 2007, Biochemistry) also suggest variable side-chain interactions in fibrils. Taken together with the 15 crystal structures of hIAPP protein segments, we surmise that there is some flexibility of side- chain interactions in the fibril.

Nonetheless, the extensive size of the 19-29 S20G dry interface implies it is particularly stable. In comparison, the zippers in our earlier publications have fewer residues and hence, smaller interfaces.